# Analysis of genome and methylation changes in Chinese indigenous chickens over time provides insight into species conservation

Tao Zeng [1,10], Jianmei Yin[2,3,10], Peishi Feng[4], Feiran Han[5], Yong Tian[1], Yuntong Wang[6], Tiantian Gu[1], Yuhui Xu[5], Yali Liu[7], Guohui Li[2,3], Liang Qu[2,3], Li Chen[1], Lihong Gu[8], Wenwu Xu[1], Qian Xue[2,3], Qingyu Wei[9], Yongqing Cao[1], Peifeng Li[9], Huiyong Zhang[2,3], Guoqin Li[1], Lijun Liu[7], Chenghao Zhou[2,3], Zhengrong Tao[1], Junda Shen[1], Wei Han [2,3✉] & Lizhi Lu [1✉]

Conservation of natural resources is a vital and challenging task. Numerous animal genetic resources have been effectively conserved worldwide. However, the effectiveness of conservation programmes and the variation information of species have rarely been evaluated. Here, we performed whole-genome and whole-genome bisulfite sequencing of 90 Chinese indigenous chickens, which belonged to the Tibetan, Wenchang and Bian chicken breeds, and have been conserved under different conservation programmes. We observed that low genetic diversity and high DNA methylation variation occurs during ex situ in vivo conservation, while higher genetic diversity and differentiation occurs during in situ conservation. Further analyses revealed that most DNA methylation signatures are unique within ex situ in vivo conservation. Moreover, a high proportion of differentially methylated regions is found in genomic selection regions, suggesting a link between the effects of genomic variation and DNA methylation. Altogether our findings provide valuable information about genetic and DNA methylation variations during different conservation programmes, and hold practical relevance for species conservation.

[1] State Key Laboratory for Managing Biotic and Chemical Threats to the Quality and Safety of Agro-Products, Institute of Animal Husbandry and Veterinary Science, Zhejiang Academy of Agricultural Sciences, Hangzhou 310021, China. [2] National Chickens Genetic Resources, Jiangsu Institute of Poultry Science, Yangzhou 225125, China. [3] Technology Innovation Co., Ltd., Jiangsu Institute of Poultry Science, Yangzhou 211412, China. [4] Zhejiang University of Technology, Hangzhou 310014, China. [5] LC-Bio Technologies CO., LTD., Hangzhou 310000, China. [6] Technical Department, Biomarker Technologies Corporation, Beijing 101300, China. [7] Zhejiang Animal Husbandry Techniques Extension and Livestock and Poultry Monitoring Station, Hangzhou 310020, China. [8] Institute of Animal Science & Veterinary medicine, Hainan Academy of Agricultural Science, Haikou 571100, China. [9] College of Animal Science, Shanxi Agricultural University (Shanxi Academy of Agricultural Sciences), Taiyuan 030032, China. [10] These authors contributed equally: Tao Zeng, Jianmei Yin.
✉email: hanwei830@163.com; lulizhibox@163.com

Animal genetic resources are the foundation of sustainable development of animal production industry, and are vital to food security and livelihoods of millions of people. Chicken (*Gallus gallus domesticus* or *Gallus domesticus*) is the most common domestic animal worldwide[1]. Nearly 1,600 different indigenous chicken breeds are internationally recognised (FAO, 2020; http://www.fao.org/poultry-production-products/production/poultry-species/chickens/en/). Yet, the value of these resources is poorly understood, and the indigenous breeds are slowly getting replaced by commercial lines. Due to this a considerable proportion of indigenous chicken breeds are becoming extinct or are at a risk of extinction, which increased from 24.75% in 2014 to 30.65% in 2018 globally[2]. Therefore, conservation of indigenous chicken breeds is important and urgent for endangered animal protection and sustainable breeding.

Currently, the conservation programmes for chickens are commonly divided into two categories: in situ conservation and ex situ in vivo conservation. In China, 128 indigenous chicken breeds have been identified and conserved at the National Conservation Farm (NCF), where the breeds evolved or are now normally found and bred (in situ)[3]. These breeds play a crucial role as a source of meat and egg, and also provide a research model for understanding the adaptations of specific breeds to specific environmental challenges, due to their various phenotypic and physiological characteristics[4]. Moreover, 28 of them have been listed in the National Conservation Catalogue and are conserved in the National Chicken Genetics Resources Gene Bank (NCGR, Jiangsu) over the past three decades (ex situ in vivo). However, for certain indigenous chicken breeds, the environment and climate in places where ex situ in vivo conservation are markedly different from those of their origins. For instance, the Tibetan chicken is a unique breed native to the Qinghai-Tibet Plateau and shows distinctive genetic adaptation to high-altitude environments[4–7]. Evaluation of the effectiveness and identification of the natural variation related to environmental adaptation during ex situ in vivo conservation of indigenous chicken breeds will further refine animal resource conservation programmes.

Since the genome of the Red Junglefowl has been sequenced and published[8], genetic basis of domestication of chicken has been widely reported using population genomics[9,10]. Recent studies have indicated that epigenetic variation plays an important role in domestication[11–14]. DNA methylation is a central epigenetic modification that plays an essential regulatory role in cellular development and environmental responses[15,16]. Natural variation for DNA methylation can represent pure epigenetic variation that occurs independently, and is not associated with any genetic variation[17]. The relationship between DNA methylation and genetic variation has been reported on a genome-wide scale in *Arabidopsis*[18]. These results provide perspectives and ideas regarding the use of genetic variation combined with DNA methylation changes to assess the differentiation of indigenous chicken breeds in different conservation programmes.

Different geographical and environmental factors have resulted in the development of divergent phenotypic characteristics and environmental adaptability of these breeds. In the present study, we performed whole-genome resequencing and whole-genome bisulfite sequencing (WGBS) of samples obtained from three phenotypically and geographically diverse Chinese indigenous chicken breeds (TC: Tibetan chicken; WC: Wenchang chicken; and BC: Bian chicken). Each breed was divided into the following three groups according to the conservation programmes: Con – cryopreserved samples, as a control, in March 2000, kept in the NCF; In – in situ conservation after ~20 years, in March 2019, kept in the NCF; and Ex – ex situ in vivo conservation after approximately 20 years, in March 2019, kept in the NCGR

(Fig. 1a and Supplementary Table 1). The aim was to evaluate the effectiveness, and investigate the genetic and DNA methylation variations occurring during different conservation programmes of three phenotypically and geographically diverse Chinese indigenous chicken breeds.

## Results and discussion

**Sample collection and sequencing**. A total of 985.74 Gb clean data were uniquely mapped to the 1.1 Gb chicken reference genome, resulting in an average of 98.44% coverage with ~9× depth for whole-genome resequencing (Supplementary Data 1 and Supplementary Fig. 1a). After SNP calling and quality-control criteria, we identified a total of 3.93 – 4.52 million high-quality SNPs and 0.64–0.75 million small insertions and deletions (InDel) for each individual. Functional annotation of the SNPs indicated that approximately 61.09% were located in introns, followed by 32.98% in intergenic regions, 3.19% in coding sequence (CDS), 2.12% in 3′-untranslated regions (3′-UTR) and 0.62% in 5'-UTR (Supplementary Data 2). For methylation, 357.20 Gb clean data were retained, which covered 71.07% of the reference genome with an average depth of 23× (Supplementary Table 2 and Supplementary Fig. 1b). After 5mC detection, a total of 91.28 % of all methyl cytosines were identified in the CG context, 1.47% in the CHG context and 7.25 % in the CHH context (Supplementary Data 3). The majority of the methylation was CG, indicating that mCG was the main type of methylation in these chicken breeds. These results were consistent with the previous studies on other chicken and pig breeds, which indicated that mCG is the major type of methylation[19,20].

**Phylogenetic analysis**. The neighbour-joining tree based on the chickens' SNPs and sequence data sets featured three groups, namely TC, WC and BC, with distinct phenotype and geographic location (Fig. 1b). These three groups were also supported by principal component analysis (PCA), which explained 7.34% of the total variation that divided TC, WC and BC into separate clusters (Fig. 1c and Supplementary Fig. 2)[21]. To estimate different ancestral proportions, we used clustering models with Admixture by assuming K ancestral populations (Fig. 1d)[22]. The cross-validation results also demonstrate clear division between TC, WC and BC at $K = 3$ (Supplementary Fig. 3). Notably, chickens under different conservation programmes basically formed two smaller branches. High consistency within each breed showed a closer genetic affinity between Ex and Con groups than that between In and Con groups (Fig. 1b), suggesting that visible genetic differentiation might occur among these three indigenous chicken breeds during in situ conservation over approximately 20 years. Additionally, two separated clades of In-TC were found in the phylogeny. Shared ancestry was also observed between few In-TC and other indigenous chickens (Fig. 1d). The results showed that the possibility that other populations could be introduced to In-TC, resulting in a visible genetic differentiation during in situ conservation of TC. Similar results were observed by Wang et al., who showed that TC may have two distinct groups, suggesting that TC could likely be traced back to two genetic sources[4].

**Genetic diversity and differentiation**. Genome-wide linkage disequilibrium (LD) in each population was estimated as the physical genomic distance at which the genotypic association ($R^2$) decays to less than half of its maximum value. Overall, the rate of LD decay was very fast in all three chicken breeds. Additionally, we observed a higher level of LD value in Ex groups of the three breeds compared with the In and Con groups (Fig. 2a and

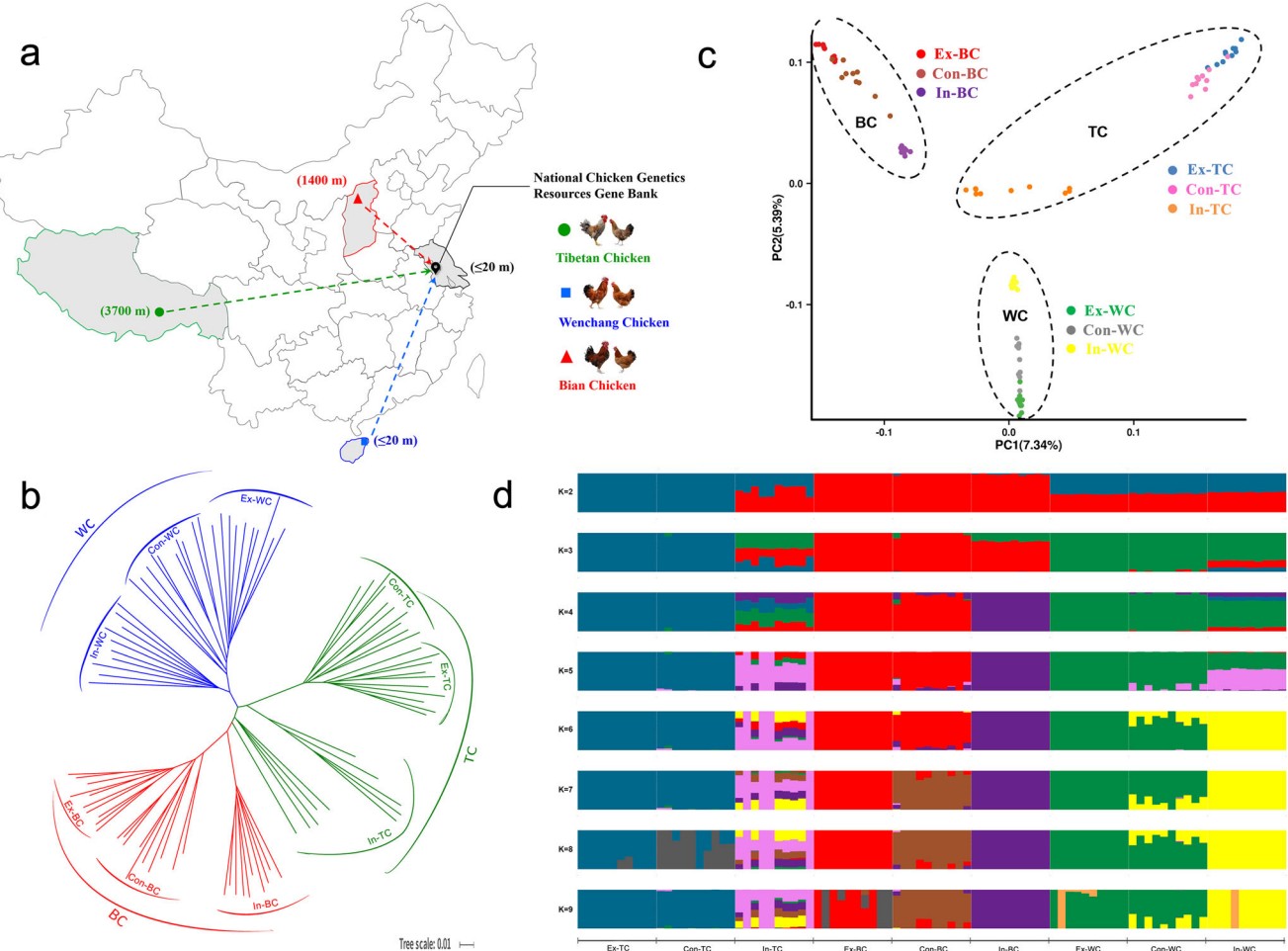

**Fig. 1 Sample distribution, phylogenetics and population structure of indigenous chicken breeds during different conservation programmes.**
**a** Geographic distribution of three selected indigenous chicken breeds. **b** The neighbour-joining tree of 90 indigenous chickens (TC: Tibetan chicken, WC: Wenchang chicken and BC: Bian chicken). Each breed was divided into three groups according to the conservation programmes (Con – cryopreserved samples, as a control, NCF; In – in situ conservation, NCF; and Ex – ex situ in vivo conservation, NCGR). **c** The principal component analysis (PCA) of 90 indigenous chickens. **d** Structure analysis of nine populations from three indigenous chicken breeds. Groupings of samples from 2–9 ancestral clusters (K) are shown.

Supplementary Fig. 4). For example, LD values for Ex-, Con- and In-TC were 246 bp, 210 bp and 216 bp, respectively. This result is consistent with a previous study, which reported that LD tends to enhanced as conservation continues under the ex situ in vivo chicken conservation[23]. The homozygosity/heterozygosity SNP ratio of different populations exhibited a high degree of consistency with the LD value, with an upward trend in the Ex groups and a decrease in the In groups when compared with Con groups (Fig. 2b).

Genetic diversity varies among species as well as within genomes, and has important implications for the conservation of species[24]. Thus, we assessed various indicators of genetic diversity under different conservation of the three indigenous chicken breeds. The average nucleotide diversity ($\pi$) and proportion of polymorphic markers ($P_N$) were lower in Ex groups of the three breeds than those in In groups and Con groups (Fig. 2c and Supplementary Table 3). However, strikingly, elevated levels of genetic diversity indicators were observed in the In groups of TC and WC breeds (Fig. 2c and Supplementary Table 3). Population genetic differentiation between the chicken breeds was measured by pairwise $F_{ST}$ values. The result was in good agreement with the phylogenetic analysis results mentioned above, which indicated that the level of $F_{ST}$ between In and Con groups was higher than

that between Ex and Con groups (Fig. 2d). Moreover, the highest $F_{ST}$ was consistently observed between the In and Ex groups of the three indigenous chicken breeds.

In conservation biology, inbreeding, or a change in allele frequencies over time, and change in different spatial scales (ranging from local to global) can cause loss of heterozygosity and genetic diversity[25,26]. Currently, ex situ in vivo conservation programme for indigenous chicken breeds are inherent to small-population paradigm, which may increase the possibility of genetic drift due to low effective population size and demographic isolation[27]. The occurrence of genetic drift can lead to a decrease in genetic diversity during ex situ in vivo conservation. We then computed Tajima's D in all populations (Supplementary Table 3). The values of Tajima's D in all groups were close to 1 (deviation from 0), indicating a possibility of weak balancing selection or a founder effect during the process of conservation. Additionally, ecological drivers, including climate change, pest outbreaks and human activities can also deeply affect genetic diversity[28–31]. A considerable number of studies have shown that species that evolved in a complex environment, such as the Qinghai–Tibetan Plateau, have comparatively low genetic diversity, for example, locusts[32], snub-nosed monkeys[33] and sheep[34]. In our study, the environment and climate of the regions of ex situ in vivo

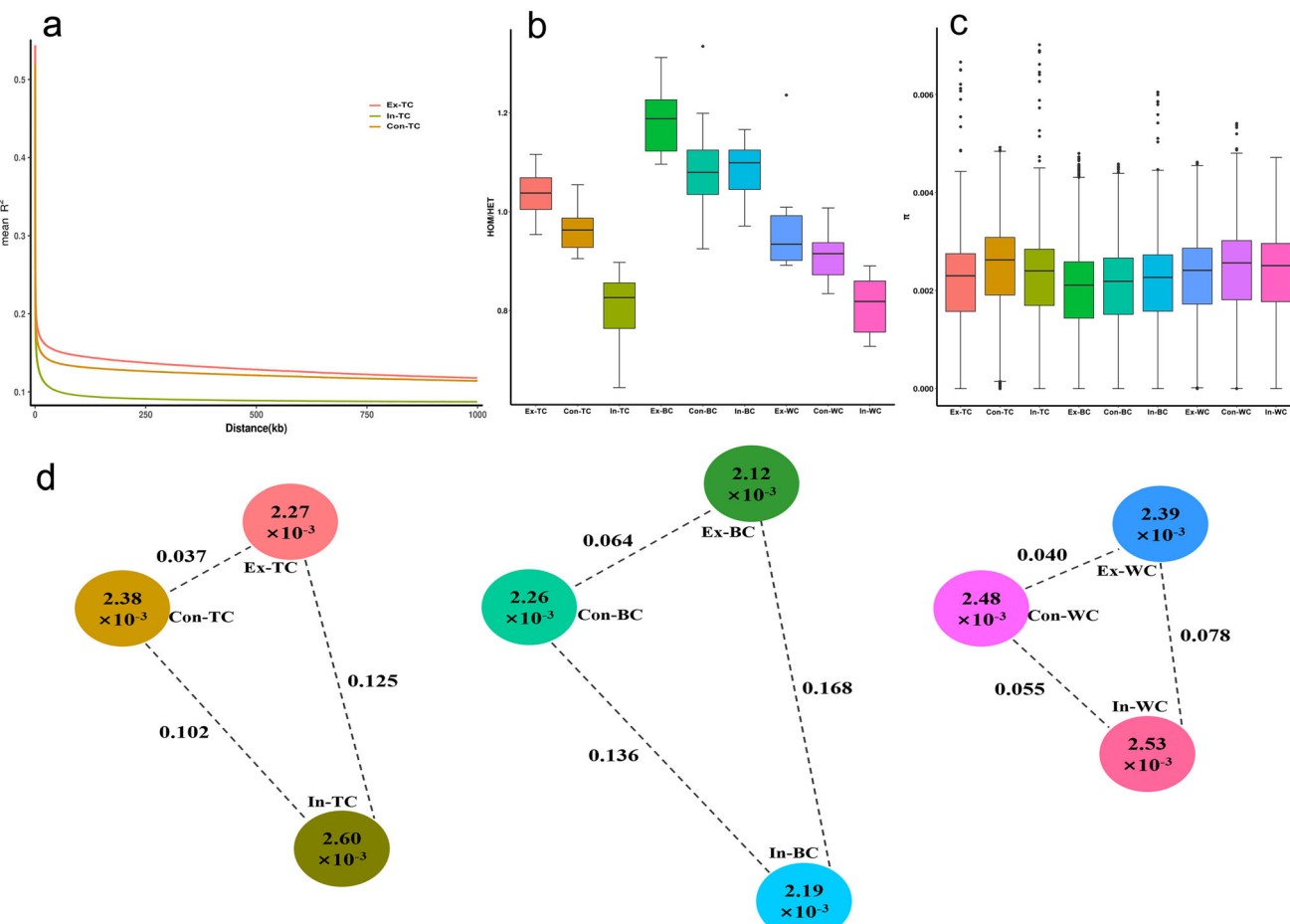

**Fig. 2 Genetic diversity and differentiation of indigenous chicken breeds during different conservation programmes. a** Patterns of linkage disequilibrium (LD) decay across the genome for TC during different conservation programmes (Con – cryopreserved samples, as a control, NCF; In – in situ conservation, NCF; and Ex – ex situ in vivo conservation, NCGR). LD decay in WC and BC are shown in Supplementary Fig. 4. X axis: physical distance between two SNPs marked in kb; Y axis: $R^2$, pearson's correlation coefficient, used to measure LD. **b** Heterozygosity across populations. Estimated as the ratio of homozygous SNPs (HOM) to heterozygous SNPs (HET) per individual. The centre line of boxplots represents median value of each population. **c** Nucleotide diversity ($\pi$) across populations. The centre line of boxplots represents median value of each population. **d** Genetic differentiation ($F_{ST}$) across populations in each breed of different conservation programmes. The value on each dotted line indicates $F_{ST}$ between the two populations, and the value in each cycle represents $\pi$ in this populations.

conservation were remarkably different from those of the origin of indigenous chicken breeds, especially for TC and WC (Supplementary Table 1). Therefore, short-term adaptability to environmental changes may result in the occurrence of adaptive selection, which can cause a decline in genetic diversity. In contrast to ex situ in vivo conservation, chickens that were subjected to in situ conservation exhibited higher genetic diversity and differentiation, especially in TC and WC. In fact, during in situ conservation, introduction of other populations of the same breed into the conservation population happens occasionally. However, low genetic diversity of BC was also observed during in situ conservation. This could be explained by the fact that BC has been considered as an endangered breed by the Chinese government, because of a rapid decrease in its population.

Overall, low genetic diversity occurred during ex situ in vivo conservation, while high genetic diversity and differentiation occurred during in situ conservation. Consequently, suitable conservation programmes should be formulated according to the physiological characteristics, living environment and risk status of indigenous chicken resources as follows: 1) In addition to NCF, the Local Genetics Resources Gene Bank (e.g., provincial level, in situ conservation) should be established to conserve indigenous

chicken resources whose living environment and climate are markedly different from the NCGR, such as Tibetan chicken. 2) Endangered chicken resources, such as Bian chicken, should be conserved in NCGR to fullest extent because of their limited population. 3) A combinative conservation programme of in situ and ex situ in vivo, and regular blood updates are recommended for normal chicken resources to maintain high levels of genetic diversity in the long term.

**Selective signatures for high-altitude adaption in TC.** Tibetans, as well as Tibetan wild and domesticated animals have revealed a highly hypoxic adaptation mechanisms to survive hypoxia occurring at high-altitudes[35–38]. To investigate the adaptive mechanisms of TC at high-altitude, we scanned the genome for regions with extreme divergence $F_{ST}$ and the highest differences in $\pi$ in 100 kb sliding windows relative to lowland populations (WC and BC). Previously, visible differentiation was observed in chickens under in situ conservation (Fig. 1b). To avoid this interference, Con group of each breed was used for selective sweep analysis. In total, we identified 228 candidate divergent regions (CDRs) ($F_{ST} \geq 0.35$) comprising 1022 candidate genes

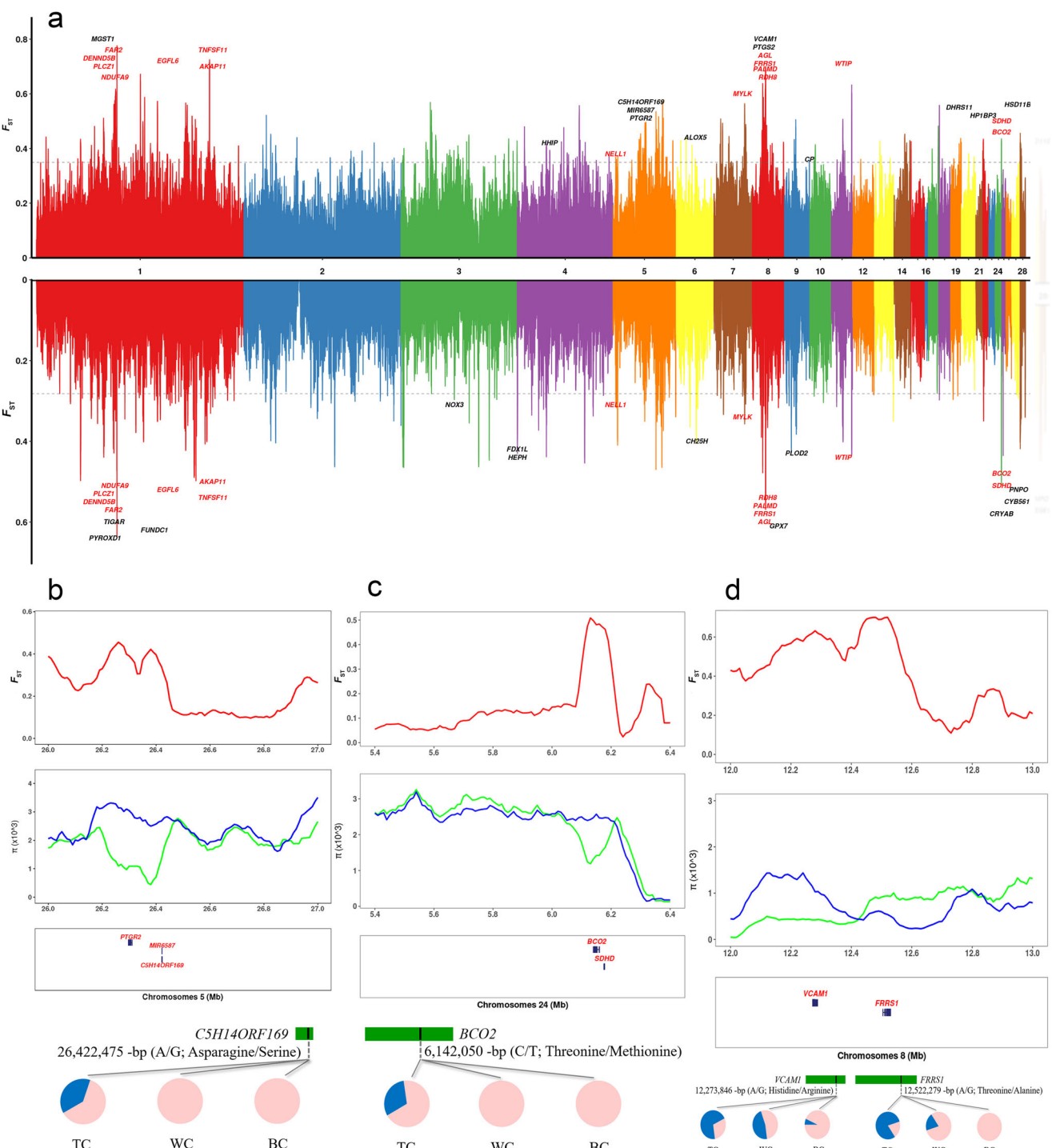

**Fig. 3 Selection signals for adaptation to high-altitude hypoxia of TC. a** Pairwise fixation index ($F_{ST}$) in 100-kb sliding windows across autosomes between TC and other chickens (upward coordinate axis: TC/BC; downward coordinate axis: TC/WC). The black horizontal dashed line corresponds to the genome-wide significance threshold ($F_{ST} = 0.35$ and $0.28$ for TC/BC and TC/WC, respectively). Genes located in divergence regions and annotated by KEGG are indicated by their gene names. Overlapping candidate genes in two selected regions are marked in red. **b–d** Three extreme CDRs, which are found close to the genes in the pathways of "oxidation-reduction process" were enriched for sites on chromosomes 5 (**b**), 24 (**c**), and 8 (**d**). The pie charts represent the spectrum of allele frequencies at the non-synonymous loci of the focused genes *C5H14ORF169*, *BCO2*, *VCAM1* and *FRRS1* in TC and other chicken breeds. The variant allele is indicated in blue, whereas the reference allele in pink.

between TC and BC (Fig. 3a and Supplementary Data 4) and 168 CDRs ($F_{ST} \geq 0.28$) comprising 700 candidate genes between TC and WC (Fig. 3a and Supplementary Data 5). Gene Ontology (GO) analysis revealed that several categories were significantly enriched for signals of selection (Supplementary Data 6–8), such

as protein kinase A signalling and motor activity, which is related to metabolic regulation. Kyoto Encyclopedia of Genes and Genomes (KEGG) pathway analysis revealed that a large number of genes were enriched into environmental information processing (Supplementary Fig. 5).

**Table 1 Summary of candidate genes that may have a role in the high-altitude adaptation of TC.**

| Symbol | $F_{ST}$ value | | GO_Term | Biological category |
|---|---|---|---|---|
| | Con-TC/Con-BC | Con-TC/Con-WC | | |
| FAR2 | 0.46 | 0.42 | GO:0055114 | Oxidation-reduction process |
| NDUFA9 | 0.62 | 0.43 | GO:0055114 | Oxidation-reduction process |
| FRRS1 | 0.70 | 0.57 | GO:0055114 | Oxidation-reduction process |
| RDH8 | 0.40 | 0.35 | GO:0055114 | Oxidation-reduction process |
| BCO2 | 0.44 | 0.51 | GO:0055114 | Oxidation-reduction process |
| AKAP11 | 0.36 | 0.32 | GO:0005777 | Peroxisome |
| EGFL6 | 0.38 | 0.28 | GO:0005509 | Calcium ion binding |
| NELL1 | 0.37 | 0.31 | GO:0005509 | Calcium ion binding |
| TNFSF11 | 0.36 | 0.32 | GO:0055074 | Calcium ion homeostasis |
| DENND5B | 0.37 | 0.30 | GO:0005262 | Calcium channel activity |
| PLCZ1 | 0.51 | 0.44 | GO:0006816 | Calcium ion transport |
| MYLK | 0.37 | 0.29 | GO:0051928 | Positive regulation of calcium ion transport |
| WTIP | 0.51 | 0.40 | GO:0001666 | Response to hypoxia |
| SDHD | 0.44 | 0.51 | GO:0071456 | Cellular response to hypoxia |

Among the selected genes, 308 genes were under selection in all TCs. Further examination of these genes revealed that 14 genes, including *FAR2*, *NDUFA9*, *FRRS1*, *RDH8*, *BCO2*, *AKAP11*, *EGFL6*, *NELL1*, *TNFSF11*, *DENND5B*, *PLCZ1*, *MYLK*, *WTIP* and *SDHD*, were functionally associated with oxidation-reduction process, calcium ion binding/transport and response to hypoxia, which indicates a role in high-altitude adaptation of the TC (Table 1 and Fig. 3a)[39–52]. Moreover, three extreme CDRs were found close to the genes associated with the pathways of "oxidation-reduction process" (Fig. 3b–d). We then identified non-synonymous SNP mutations in selected genes of extreme CDRs and found that the variant allele frequencies of non-synonymous SNPs in four genes (*C5H14ORF169*, *BCO2*, *VCAM1* and *FRRS1*) were significantly different ($P < 0.01$) between TC and other chicken breeds (Fig. 3b–d).

**Differentially methylated regions in chickens between in situ and ex situ in vivo conservation programmes.** To elucidate the DNA methylation variations occurring during different conservation programmes of the three indigenous chicken breeds, we identified differentially methylated regions (DMRs) between different populations according to a previously described method[18,53]. Compared with Con group, 4987 DMRs were detected in Ex group of TC, including 4582 CG-DMRs, 1 CHG-DMR and 404 CHH-DMRs (Fig. 4a and Supplementary Data 9). While, compared with Con group, 3833 DMRs were identified in In group, which included 3362 CG-DMRs, 1 CHG-DMR and 470 CHH-DMRs (Supplementary Data 10). Similar trend was observed in WC with a total of 6394 DMRs (6143 CG-DMRs, 1 CHG-DMR and 250 CHH-DMRs) in Ex group and 3957 DMRs (3638 CG-DMRs, 1 CHG-DMR and 318 CHH-DMRs) in In group (Fig. 4a and Supplementary Data 11–12). In contrast, compared with Ex group of BC (3864 CG-DMRs, 69 CHG-DMRs and 223 CHH-DMRs), more DMRs were identified in In group (5360 CG-DMRs, 3 CHG-DMRs and 450 CHH-DMRs; Supplementary Data 13–14).

Then, we identified DMRs between Ex groups and In groups of the three indigenous chicken breeds. The results showed that DMRs between Ex and In groups of TC, WC and BC increased by 39.47%, 13.57% and 2.32%, respectively, when compared to DMRs between In and Con groups (Fig. 4b). The results further indicated that methylation difference between Ex and In groups were higher in TC than other breeds. Subsequently, we compared the characteristics of DMRs in different contexts, and found that CHH-DMRs were significantly different from CG-DMRs. A higher proportion of CHH-DMRs was found in genic regions, whereas CG-DMRs were mainly located in intron regions

(Supplementary Fig. 6). In addition, we found that the average length of CHH-DMRs was less than that of CG-DMRs. However, no significant difference in the length of CHH-DMRs and CG-DMRs were observed between In and Ex groups (Fig. 4c). To investigate the potential functions of the differentially methylated genes between In and Ex groups in chickens, functional enrichment analyses of the genes with DMRs were performed based on their annotation in GO and KEGG databases. The results showed that many DMRs were enriched in signal transduction, metal ion binding and transcription regulation activity. KEGG pathway analysis showed that the differentially methylated genes were mainly enriched in the neuroactive ligand-receptor interaction, MAPK signalling pathway and calcium signalling pathway (Supplementary Figs. 7–9).

DNA methylation provides a mechanism for organisms to adapt, within and between generations[54]. Numerous studies have indicated that DNA methylation can be responsive to climate change and plays an important role in certain developmental processes[13,55]. In our analysis, more DMRs were found in Ex group of TC and WC. Given the variety of DMR length, we identified overlapping bases (bp) and found that only a small proportion of these were found in the DMRs between different conservation programmes (Fig. 4d). This indicates that different DNA methylation-mediated adaptation mechanisms existed in the three chicken breeds. Further investigation showed that relatively few bases (7.41% and 7.83%) in the DMRs showed overlapping in Ex group in comparison with In group (9.89% and 13.15%) in TC and WC, respectively. However, we observed an opposite trend for different conservation programmes in BC (Fig. 4d). DNA methylation variation levels were higher in Ex group of TC and WC than their respective In group. This could be explained by the differences in climatic and geographic features of the two conservation types. Interestingly, combined with previous analyses, we found an opposite trend in DNA methylation variation levels and genetic diversity (Fig. 2c), suggesting that an association might exist between these two variations. Furthermore, we found that a few bases in the DMRs were overlapped in the Ex group of the three chicken breeds (Fig. 4e).

**DNA methylation variations contribute to genomic selection signatures during Ex conservation programme of TC.** DNA methylation variations can lead to extensive phenotypic variations, such as environmental adaptation[18,56], energy use efficiency[57], and disease resistance[58]. It has been proposed that methylation plays a crucial role in cellular response to hypoxia[59]. Meanwhile, recent studies have indicated that local genetic variations can influence DNA methylation[18,60,61]. To identify the

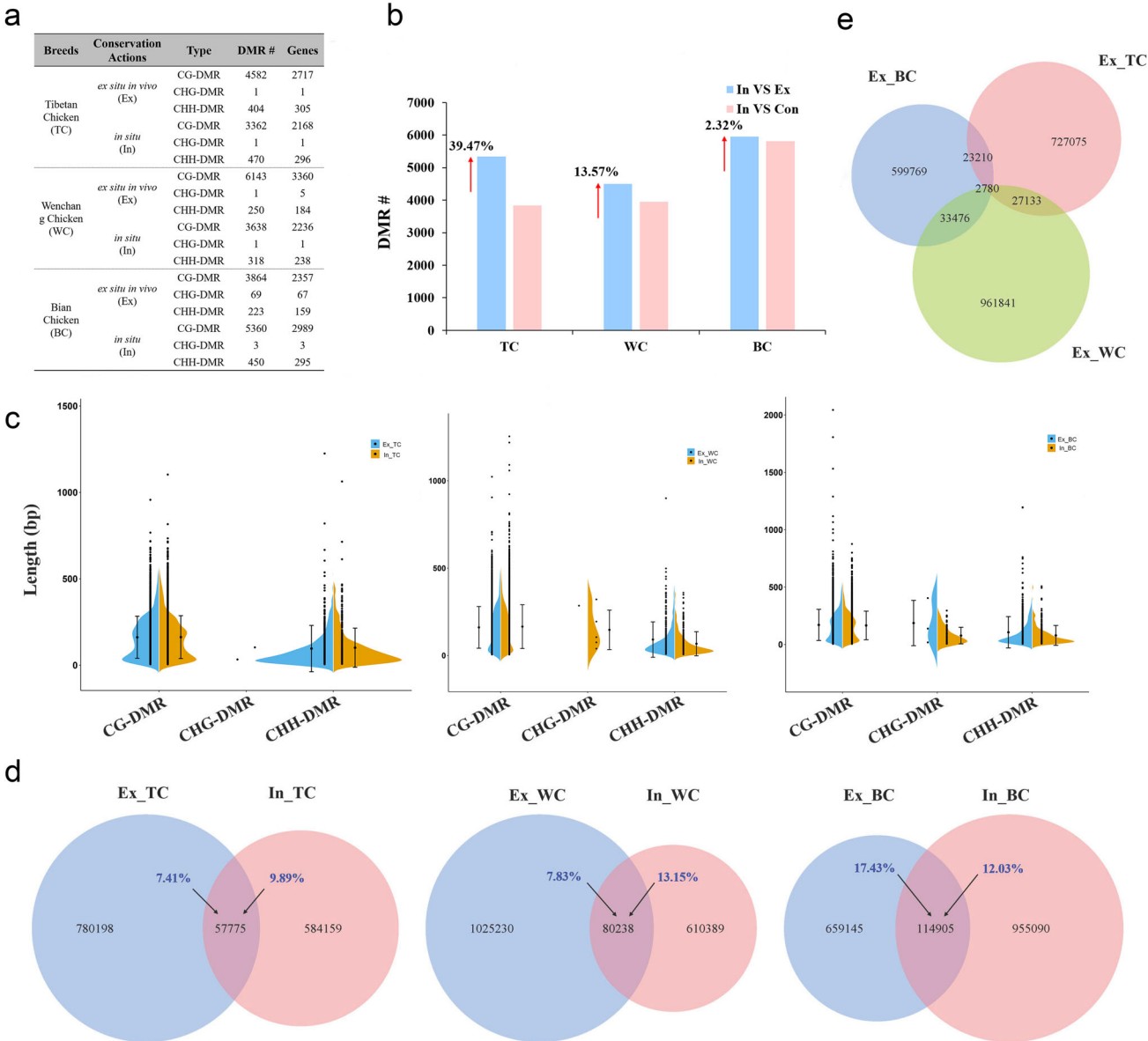

**Fig. 4 Differentially methylated regions (DMRs) during in situ (In) and ex situ in vivo (Ex) conservation programmes of chickens. a** Number of DMRs detected in different conservation programmes of three indigenous chicken breeds. **b** Change in numbers of DMRs detected between Ex and In of three indigenous chicken breeds. **c** Length of DMRs during different conservation programmes of three indigenous chicken breeds. **d** Venn diagram of the numbers of overlapping bases in DMRs during different conservation programmes of three indigenous chicken breeds. Overlapping areas are presented as percentages. **e** Overlapping bases (bp) in DMRs during Ex conservation of three indigenous chicken breeds.

DMRs that might be associated with local genetic variations, we investigated the percentage of DMRs in $F_{ST}$ selection regions and other regions via whole genome scanning. Overall, a high proportion of DMRs was found in both the $F_{ST}$ and non-$F_{ST}$ regions, during Ex conservation programme of the three chicken breeds (Fig. 5a). Meanwhile, we found that the proportion of DMRs was higher in $F_{ST}$ selection regions than in non-$F_{ST}$ regions in the Ex of TC, whereas approximately the same in group of TC, and Ex and In groups of WC (Fig. 5a). In contrast, the proportion of DMRs was higher in non-$F_{ST}$ regions in Ex of BC. Taken together, these results indicate that the variation patterns, both genetic and DNA methylation, were different during in situ and ex situ in vivo conservation programmes in chickens, which could relate to the features of the environment of origin, such as altitude and temperature, risk status and so on. Remarkably, the high proportion of DMRs in the $F_{ST}$ regions of Ex group of TC suggests a

link between these DMRs and local genetic variations. It is well known that the climatic and geographic characteristics of ex situ in vivo conservation of TCs are markedly different from those of in situ conservation. Therefore, local genetic variations due to environmental differences in the Ex group of TC may influence DNA methylation level, and ultimately lead to changes in gene expression. In our analyses, the DMRs that were found in the $F_{ST}$ regions were considered as "sweep DMRs (s-DMRs)".

In total, 334 s-DMRs comprising 323 s-CG-DMRs and 11 s-CHH-DMRs were identified in the Ex group of TC (Supplementary Data 15). The s-DMRs contained 124 candidate genes. Further examination of these genes identified eight genes, including *HTR2A*, *EYS*, *DST*, *RAP1GDS1*, *HTT*, *HTR7L*, *FKBP7*, and *MYLK*, that have functional associations with calcium ion binding/calcium signalling pathway, suggesting potential roles for these genes in high-altitude adaptation of TC (Fig. 5b and

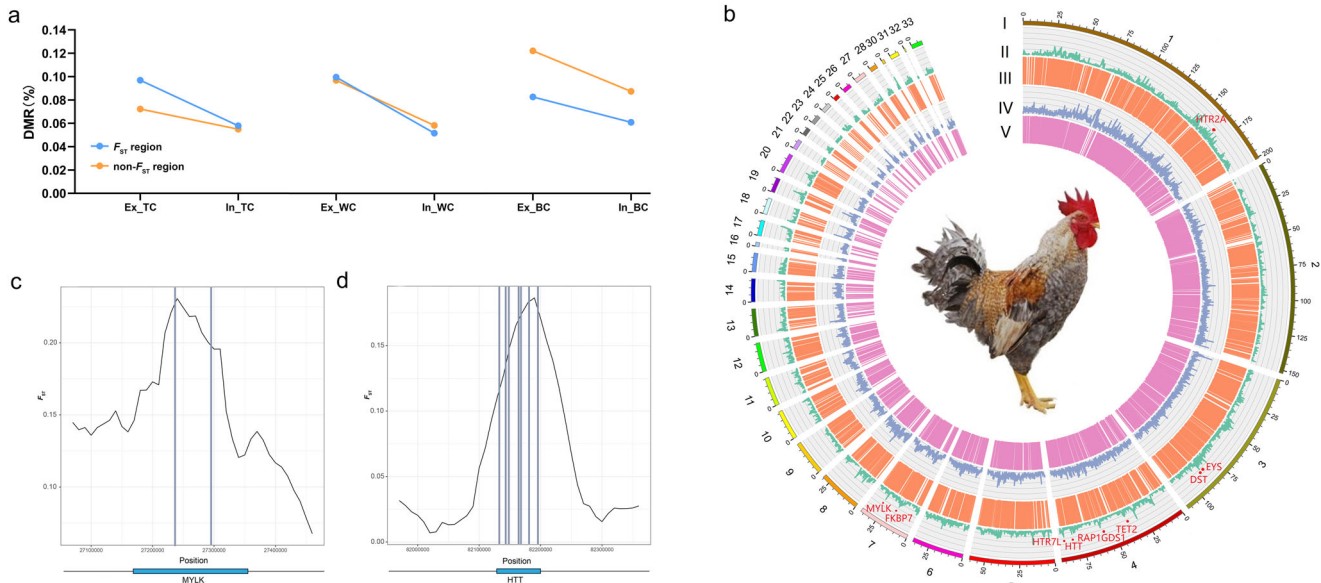

**Fig. 5 DNA methylation variations contribute to genomic selection signatures (s-DMRs) during Ex conservation programme of TC. a** Percentage of DMRs in $F_{ST}$ selection regions and other regions of indigenous chicken breeds (TC: Tibetan chicken, WC: Wenchang chicken and BC: Bian chicken) during different conservation programmes (In – in situ conservation and Ex – ex situ in vivo conservation). **b** Genome and methylation landscapes during different conservation programmes of TC. From outer to inner circles are as follows: chromosome scheme (I), selective sweeps detected during Ex programme (II), DMRs during Ex programme (III), selective sweeps detected during In programme (IV), and DMRs during In programme (V). Red dots in circle II denote genes located in s-DMRs. **c**, **d** Two prominent genes *MYLK* (**c**) and *HTT* (**d**), which were located in s-DMRs during Ex programme of Tibetan chickens. Grey shadow depicts the location of DMRs.

Supplementary Data 16). $Ca^{2+}$ is one of the most important regulators of pulmonary vascular function. Calcium signalling stimulates the translation of HIF-1, a principal regulator of the transcriptional response to hypoxia[62–65]. One prominent candidate gene, *MYLK*, which encodes a $Ca^{2+}$/calmodulin dependent enzyme, was identified in s-CG-DMRs (Fig. 5c) and as well as in CDRs between TC and other chickens (Fig. 3a and Table 1). Previous studies have reported that MYLK protein levels are up-regulated through the mediation of HIF-1α with concomitant vascular endothelial barrier dysfunction when the cells are exposed to hypoxic stress[66]. *MYLK* upregulation is associated with augmented vasoconstriction, a marker of high-altitude pulmonary oedema[67]. Interestingly, recent studies have suggested that hypoxia upregulates the expression of *MYLK*, and thereby increases the potential to transport oxygen by promoting smooth muscle contraction in TC embryos[68]. We also highlighted two candidate genes, *HTT* and *EYS*, which contained seven DMRs and one DMR in the selection region, respectively (Fig. 5d and Supplementary Fig. 10a). Although the role of *HTT* gene in high-altitude hypoxia adaptation is unknown, previous studies have found an association between *HTT* and InsP₃R1-mediated neuronal $Ca^{2+}$ signalling, and have provided an explanation for the derangement of cytosolic $Ca^{2+}$ signalling in Huntington's disease patients and mouse models[69]. Furthermore, *TET2*, which is a key factor in DNA demethylation, and serves as a methyl cytosine dioxygenase, was found in s-CG-DMRs (Supplementary Fig. 10b). Hypoxia has been shown to influence DNA methylation either by transcriptionally activating *TET* or by reducing *TET* activity[70,71]. Recently, *TET* has also been shown to affect hypoxia signalling through its interaction with HIF-1α[72,73]. Overall, these genes may be associated with the high-altitude adaptation in TC.

## Conclusions

Conservation of animal genetic resources plays an important role in biodiversity protection. In the present study, we performed whole-genome sequence analysis and WGBS analysis of three indigenous chicken breeds that have been conserved in different conservation programmes. Our results demonstrated that a higher genetic diversity and differentiation occurs during in situ conservation, while reduced genetic diversity occurs during ex situ in vivo conservation programme of three indigenous chicken breeds. DNA methylation variation levels were higher during ex situ in vivo conservation were higher than during in situ conservation. Moreover, a high proportion of DMRs in genomic selection regions was found during ex situ in vivo conservation of Tibetan chickens, suggesting an association between these DMRs and local genetic variations. Additionally, we discovered several candidate genes in selective sweep regions and DMRs, which correlated with calcium signalling pathway, suggesting a potential regulatory mechanism underlying high-altitude adaptation in Tibetan chickens. Collectively, our study provides valuable information about the genetic variations and DNA methylation variations during different conservation programmes of indigenous chicken breeds, contributes to a profound understanding of high-altitude adaptation and will facilitate future endangered species conservation.

## Methods

**Sampling, DNA sequencing and WGBS.** Three Chinese indigenous chicken breeds that were conserved in different conservation programmes (in situ and ex situ in vivo) were used in this study (TC: Tibetan chicken; WC: Wenchang chicken; and BC: Bian chicken). Each breed was divided into the following three groups according to the conservation programmes (Con – cryopreserved samples, as a control, in March 2000, kept in the NCF; In – in situ conservation after approximately 20 years, in March 2019, kept in the NCF; and Ex – ex situ in vivo conservation after ~20 years, in March 2019, kept in the NCGR) (Fig. 1a and Supplementary Table 1). Blood samples were collected from a total of 90 individuals, of which 30 for each breed and 10 for each group in the same breed. Animals used in this study were raised in accordance with the national standard of Laboratory Animal Guidelines for ethical review of animal welfare. All experiment procedures were approved by the Animal Use Committee of Zhejiang Academy of Agricultural Sciences (No. 20-022).

Genome DNA was extracted using a DNA isolation kit (Tiangen, Beijing, China) according to the manufacturer's instructions. For genome sequencing, a

minimum of 0.5 µg of genomic DNA from each sample was used to construct a library with an insert size of ~350 bp. Paired-end (PE) sequencing libraries were constructed according to the manufacturer's instructions (Illumina Inc., San Diego, CA, USA) and sequenced using Illumina HiSeq 4000 platform (Illumina, San Diego, CA, USA). WGBS libraries were prepared according to the protocol described in a previous report[74]. The DNA samples were fragmented with sonication and subjected to bisulfite conversion. Ultra-high-throughput paired-end sequencing was carried out using Illumina HiSeq 4000 platform (Illumina, San Diego, CA, USA) according to the manufacturer instructions. Raw HiSeq sequencing data were processed by Illumina base-calling pipeline (SolexaPipeline-1.0).

**Reads mapping and variant calling**. To obtain high-quality data, we implemented quality-control procedures to remove the following types of reads: (1) read pairs containing adapters, (2) reads with ≥ 10% unidentified nucleotides (N), and (3) >50% of the read bases with a Phred quality score (i.e., $Q$-score) <10. A total of 985.74 Gb of high-quality paired-end reads were generated with an average quality of 96.98% for Q20 and 93.66% for Q30. The filtered high-quality data were mapped to the chicken reference genome (ftp://ftp.ncbi.nlm.nih.gov/genomes/all/GCF/000/002/315/GCF_000002315.5_GRCg6a/GCF_000002315.5_GRCg6a_genomic.fna.gz) using Burrows Wheeler Aligner (v0.7.8) with default parameters[75]. Mapping results were then converted into the BAM format and sorted using SAMtools (v1.2). On an average, 98.44% of the reads were mapped, resulting in a final average sequencing coverage of 9× per individual. The mapped reads were further filtered by removing duplicate reads using the Picard MarkDuplicates (http://sourceforge.net/projects/picard/). After alignment, we performed SNP calling using GATK (v3.7), and the output was further filtered using VCFtools[76,77]. Moreover, to exclude SNP callings errors, variant sites with QD < 2.0, QUAL < 30, MQ < 40, FS > 60.0 were discarded. The process of indel calling was the same as described for the SNPs. Heterozygosity and homozygosity for each sample were calculated by the total number of callable sites across the whole genome. Finally, SNP variations were annotated using the SnpEff software (v4.3)[78]. The whole-genome sequencing data and distribution of SNPs and InDels in the three Chinese indigenous chicken breeds are shown in Supplementary Data 1–2.

**Phylogenetic tree, population structure, principal component analysis and Admixture**. To determine the phylogenetic relationships, we constructed a neighbour-joining phylogenetic tree based on MEGA X software (v10.1.5) with Kimura 2-parameter model and 1000 bootstrap replicates[79]. Population structure was then inferred using the software ADMIXTURE (v1.3.0) to quantify genome-wide admixture between three Chinese indigenous chicken breeds populations[22]. The number of genetic clusters $K$ ranged from 2 to 9 with 10,000 maximum iterations. In order to understand the relationships between different geographic populations and same breed with different conservation actions, principal component analysis (PCA) of the 90 samples was performed based on all SNPs using EIGENSOFT software (v7.2.1)[21].

**Genetic diversity, differentiation and linkage disequilibrium**. The inbreeding coefficient ($F$), observed heterozygosity ($Ho$), expected heterozygosity ($He$) and proportion of polymorphic markers ($Pn$) were calculated using PLINK (v1.07)[80]. Effective population size ($Ne$) for each breed was calculated using default parameters in N$_e$ESTIMATOR software (v1.3)[81]. The nucleotide diversity ($\pi$) and Tajima's D ($\theta_w$) were calculated based on the list of high-quality SNPs using VCFtools (v0.1.14) with a sliding window approach (100-kb windows with 10-kb steps)[82]. Population differentiation was measured by pairwise $F_{ST}$ using the unbiased estimator of Weir and Cockerham (1984) with default parameters[83]. To evaluate linkage disequilibrium (LD) for different groups, we calculated the correlation coefficient ($r^2$) between alleles at two separate SNP loci using Haploview (v4.2) with parameters "-maxdistance 500 -dprime -minMAF 0.05 -hwcutoff 0.001"[84].

**Genome scanning of selective signatures**. To identify genome-wide selective sweeps associated with high-altitude adaptation. We screened the chicken genomes with a sliding window approach (100-kb windows with 10-kb step length), and estimated $F_{ST}$ and $\pi$ ln ratio values for each window using VCFtools (v0.1.14) and in-house scripts for comparison between TC and BC and between TC and WC[77]. The significance threshold was set to the top 5% for $F_{ST}$ and $\pi$ ln ratio. We then considered the windows with top 5% simultaneously as candidate selective regions under strong selective sweep and subsequently examined for potential candidate genes. Finally, we identified 228 candidate divergent regions (CDRs) ($F_{ST} \geq 0.35$) comprising 1022 candidate genes between TC and BC (Fig. 3a and Supplementary Data 4) and 168 CDRs ($F_{ST} \geq 0.28$) comprising 700 candidate genes between TC and WC (Fig. 3a, Supplementary Data 5). Functional enrichment terms, including Gene Ontology (GO)[85] and KEGG pathways[86] were retrieved for these genes. GO terms and pathways with enrichment q-values < 0.05 were considered to be significantly enriched.

**WGBS analysis**. Adapters and low-quality bases in the WGBS reads were removed using in-house Perl scripts. The sequence quality was then verified with FastQC (http://www.bioinformatics.babraham.ac.uk/projects/fastqc/). A total of 357.20 Gb of clean reads was generated with an average quality of 91.93% for Q30. Reads that passed quality control were mapped to the chicken reference genome by Bismark (v0.21.0)[87]. After filtering the duplicate reads, the methylation information for each cytosine site was extracted. Methylation states were evaluated based on the binomial distribution test principle followed by coverage ≥ 4X and false discovery rate (FDR < 0.05) correction. The methylation level (CG, CHG and CHH) was defined as the ratio between the number of reads of a methylated C to the total number of reads of that site[88].

**DMR detection and functional enrichment analysis**. Genome coverage of CG, CHG and CHH sites under different sequencing depths, and distribution of clean reads in different CG density regions were analysed. The genome coverage of the CG, CHG and CHH sites in every chromosome and different genome components including promoter, gene body and downstream was also analysed. DMRs were detected according to the cytosine (C) mapping of uniquely mapped reads into the reference genome. MOABS (v1.3.7.7) was used to screen DMRs, and the coverage depth was required to be no less than 10X, with at least three different methylation sites[89]. DNA methylation levels of different groups were compared pairwise through Fisher's exact test and the p-values were adjusted for multiple comparisons using the Benjamini-Hochberg method. CG-DMR candidate regions, CHG-DMR candidate regions, and CHH-DMR candidate regions were required to have an average methylation level differences of >0.3, >0.2 and >0.2, respectively, between the corresponding populations. Finally, the regions with an FDR less than 0.05 were identified as DMRs. We identified DMRs between Con and Ex, Con and In groups and between Ex and In groups of the three indigenous chicken breeds. To explore the function of the genes containing DMRs, GO enrichment and KEGG pathway analyses were conducted for differentially methylated genes using Blast2GO (v5.2.5)[90]. GO terms and pathways with enrichment q-values <0.05 were considered to be significantly enriched.

**Statistics and reproducibility**. Details about experimental design and statistics used in different data analyses are given in the respective sections of results and methods. Unless stated otherwise, statistical analysis was performed using base and nlme in R(4.2.0). Details allowing the reproducibility of all analyses are provided in the methods section. For genome resequencing, we used 90 chickens of different conservation programmes of three Chinese indigenous chicken breeds. For whole-genome bisulphite sequencing, we used at least three chickens samples for pool from nine subgroup of three Chinese indigenous chicken breeds.

**Reporting summary**. Further information on research design is available in the Nature Research Reporting Summary linked to this article.

## Data availability

All raw sequencing reads have been deposited in the National Center for Biotechnology Information (NCBI) Sequence Read Archive database (https://www.ncbi.nlm.nih.gov/sra) under project PRJNA698651 and PRJNA699090. The numerical source data for graphs are available in Supplementary Data 17.

## Code availability

Details of publicly available software used in the study are given in the Methods. The R code is available in Rcode.R (Supplementary Software 1).

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

## Acknowledgements

We thank many people not listed as authors who provided feedback, samples, and encouragement, especially Jian Xu, Xue Du, Linde Wu, Jin Wang, Wang Shi, and Chunlan Wang. This work was supported by National Key Research and Development Program of China (2021YFD1200302), Project of Key Research and Development Plan (Modern Agriculture) of Jiangsu Province (BE2019353), "JBGS" Project of Seed Industry Revitalisation in Jiangsu Province (JBGS〔2021〕029) and Key Research and Development Program of Zhejiang Province (2021C02034).

## Author contributions

T.Z., L.Z.L. and W.H. planned the project. L.Z.L., W.H., T.Z., J.M.Y., P.S.F. and Y.T. designed the research. T.Z., J.M.Y., T.T.G., Y.L.L., G.H.L., L.Q., L.C., L.H.G., Q.X., Q.Y.W., Y.Q.C., P.F.L., H.Y.Z., G.Q.L., L.J.L., C.H.Z., Z.R.T., J.D.S. and W.H. prepared the research materials. T.Z., P.S.F., F.R.H., Y.T.W., Y.H.X., Y.T. and W.W.X. performed the experiments for sequencing and the data analyses. T.Z. drafted the manuscript. T.Z., L.Z.L. and W.H. revised the manuscript. All authors read and approved the final manuscript.

## Competing interests

The authors declare no competing interests.
