## [Peer Review File · Communications Biology]

Reviewers' comments:

Reviewer #1 (Remarks to the Author):

The paper evaluates genetic diversity issues of three chicken breeds from different geographic locations. Analysis were performed, dealing with genotypes, sequence, and methylation. A major limitation of this paper is the relatively small number animals used in the analysis. While some breed comparisons have been done with 25 to 30 animals, to subdivide each breed into three groups of 10 animals each makes interpretation of the population genetics inconclusive. While the paper reports how the populations have drifted apart over the 20 year span of time this is not to be unexpected given the small sample size and genetic drift. That said genetic drift is an important element in conserving ex-situ in-vivo populations.

The paper also needs to expand its description of the populations and how they are managed. A major issue is how populations management is referenced. In the paper these are described as:

- R -ex-situ in-vivo or control around the year 2000. Since this represents 20 years before the experiment I am assuming these samples were cryopreserved.
- O – in situ conservation >20 years, NCF
- A - ex-situ in-vivo >20 years, NCGR

Based upon FAO guidelines the manner in which the populations are referred to is incorrect and therefore leads to confusion in evaluating the results. If samples were collected in ~2000 they would have been cryopreserved or frozen until analyzed, therefore to refer to them as ex-situ in-vivo is incorrect. The only difference between O and A is the physical location, neither (based on Figure 1 map) is where the breeds were developed. Therefore, they both should be designated as ex-situ in-vivo. Making some assumptions about what the authors may have meant the following adjustments are suggested:

- R – Control, cryopreserved/frozen samples,
- O – ex situ in-vivo NCF
- A - ex situ in-vivo NCGR

As such what is being evaluated is how the O-NCF and A-NCGR populations have drifted apart from R.

The authors should carefully review their statements on LD as they seem to imply that an increase in LD is a positive aspect for conservation rather than a negative.

Reviewer #2 (Remarks to the Author):

This paper reports on a study comparing genomic and methylation changes that occur during conservation of specific indigenous chicken breeds in China. In particular, the authors had a specific interest in the Tibetan chicken, a unique breed native to the Qinghai-Tibet Plateau, which shows distinctive genetic adaptation to high-altitude environments. The aim of this study was to explore this further by comparing the Tibetan chicken to other low-altitude breeds and to identify candidate genomic regions/genes underlying adaptation to high-altitude. This work potentially has significance to conservation programmes and adaptation of specific to environmental changes.

The study raises several interesting observations on genomic variation of genetic and epigenetic characteristics within/between indigenous chicken breeds. Then tries to test for any association (if any) between these measures of genetic and epigenetic variation. Finally, the study explores specific regions/genes for evidence of genetic and/or environmental adaptability to high-altitude. My main concern is that this study is mostly observational and correlates the characteristics of lists of genes highlighted from enrichment analyses from multiple comparisons (between breeds, between conservation programs, etc.), looking for any evidence that supports the role of genes in adaptability towards high-altitude. I would prefer an unbiased, whole genome analysis based on robust statistics. The current study rarely demonstrates this. For example:

- There needs to be a justification of the sampling plan: 3 breeds x 3 conservation regimes x 10 samples each = 90 samples in total – is this sufficient for the comparisons made? (Power Analysis is needed).
- How robust is the NJ phylogenetic tree? How robust are the branches of the tree (e.g., based on bootstrapping resampling)?
- What was the statistical measures of significance for the various (LD, Methylation, DMRs, etc.) whole genome analyses?
- The cross-validation analysis of the Structure data was interesting with a minimum error for K= 3, but how significant was this compared to K = 2 or K = 4?
- How significant were the overlaps between regions under selective sweeps and DMRs? For example, there are 1000s of DMRs, any overlap could be a sampling artefact (i.e., chance).
- How significant were the enrichment analyses for GO and KEGG terms? Were these corrected for multiple testing?
- What is the resolution (and significance) of CGRs/DMRs? How many genes are contained within each region?

Minor but important comments:

- I think English is not the native language of the authors, so I would suggest they seek help with spelling, grammar, and layout of the paper.
- The authors frequently make statements (e.g., methylation is associated with changes in environmental conditions) but do not provide any further explanation or context.
- Finally, what are “partial candidate genes”?

Reviewer #3 (Remarks to the Author):

This is a well written paper that describes a unique data set which the authors use to assess the impact of ex-situ conservation programs on the genome and epigenome of a model species (chicken). The results are quite interesting and will be relevant for a relatively broad community. I found most of the analyses/results sound (see below for specific queries) and the overall message was quite clear. I have however a few questions for the authors:

I suggest re-writing the abstract - it does not really work right now as too many details are given in about the results but there are no hypotheses / questions for example:

Line 10-11: This sentence: " Moreover, a higher proportion of differentially methylated regions in genomic selection regions was found in Tibetan chickens, suggesting a link between the effects of genomic variation and DNA methylation." on its own, makes little sense in the abstract - why specifically Tibetan chickens?

Line 13-14: Without context this sentence " Further analyses revealed that several candidate genes were enriched in the calcium-signaling pathway, which is usually related to high-altitude adaptability " does not make much sense

line 23: this statement is misleading: this is what the paper cited says: "A molecular clock analysis suggests that domestic chickens diverged from *G. g. spadiceus* ~9500 ± 3300 years ago, though this node does not necessarily correlate with the beginning of domestication process, as chickens are archaeologically visible much later"

line 45: comparatively to what?

Generally please limit the use of acronym to a strict minimum, it is really hard to remember all these

acronyms here. I can spot 7 in a single page. Also at least please explain the acronym in the figure legend. Lastly, please use acronym that are more intuitive? Why is A = ex-situ? O = in-situ? why R= control? I am not sure these are going to save that many words? It makes it very difficult to read the figures especially.

I am not sure why the author need to add "in-vivo" to ex-situ?

Line 101: change "across" to "within" - make it clearer that animals of the same breed still cluster together.

Line 105-107: I don't think this is necessarily true - could this just be a founder / drift effect? Maybe test this hypothesis using D-statistics? I think this is a very important point here - is the ancestry of these breeds being diluted during conservation? if yes could affect some of the methylation patterns?

Line 117: The result of the increase in LD is very interesting - I would add some more quantitative results in the in text. Also, change decayed to delayed

Line 122: how is the hom/het calculated? are the authors considering hom non-ref? or hom-derived allele? or hom sites which are het in one individual? I am also not sure how this can be interpreted - why is this consistent with LD?

Line 127-128: could this be due to admixture? This would be great to test.

Line 148-149: Although I agree that some changes might be adaptive, most differences are likely to be the results of a founder / population bottleneck. the authors should generate a few more summary statistics (e.g. Tajima's D?) here which I think could be helpful to see what is going on.

Line 161-164: I think this is a bit strong worded here - it is possible that many (or even all) the differences are due to founder effect rather than adaptation.

The differentially methylated region analyses are unclear - the authors need to add more information in the method section. What are these differences? Are they between control and ex-situ/in-situ populations? Or are these loci that are differentially methylated within each population? If the latter, are the authors saying that there are more DMRs within ex-situ than in-situ populations? Why would that be? What I think would have been more interesting to look at methylation differences between in-situ and ex-situ to see whether Tibetan population for example shows more differences than other breeds?

line 249-252 - could you run a test for this maybe?

line 259-261: this is weirdly phrased sounds like the authors are saying that DNA methylation creates mutations?

Dear Reviewers,

Thank you for your letter and the comments concerning our manuscript entitled “**Genome and methylation changes in different conservation of indigenous chickens over the last 20 years**”. Those comments are all valuable and very helpful for revising and improving our paper, as well as the important guiding significance to our researches. We have studied comments carefully and have made correction which we hope meet with approval. The main corrections in the paper and the responds to the reviewer’s comments are as following:

Responds to the reviewer’s comments:

Reviewer #1 (Remarks to the Author):

The paper evaluates genetic diversity issues of three chicken breeds from different geographic locations. Analysis were performed, dealing with genotypes, sequence, and methylation.

A major limitation of this paper is the relatively small number animals used in the analysis. While some breed comparisons have been done with 25 to 30 animals, to subdivide each breed into three groups of 10 animals each makes interpretation of the population genetics inconclusive. While the paper reports how the populations have drifted apart over the 20 year span of time this is not to be unexpected given the small sample size and genetic drift. That said genetic drift is an important element in conserving ex-situ in-vivo populations.

Author response: Special thanks to you for your good suggestions. For the conservation of animal genetic resources, population size is crucially important. An efficient conservation scheme relies on an effective population size, as well as an effective selection and mating strategy. In China, the government and experts have developed a relatively complete conservation plan (*in situ* conservation and *ex situ in vivo* conservation). Briefly, the conservation goals are that population sizes should be kept constant across generations (30 males and 300 females), and random mating should be enforced within families, with one son kept per sire family and one daughter kept per dam family (Zhang, M. et al. *Genomic diversity dynamics in conserved chicken populations are revealed by genome-wide SNPs. BMC Genomics. 19, 598 (2018).*). Therefore, population size is sufficient and the selection and mating strategy are basically identical whether *in situ* conservation and *ex situ in vivo* conservation. Environmental adaptability is the main factor that determines the difference of chickens under different conservation programmes. We randomly selected 10 individuals from different populations for scientific research.

Moreover, when the chicken breed was listed in the National Conservation Catalogue and first conserved in the National Chicken Genetics Resources Gene Bank (Jiangsu, NCGR) for *ex situ in vivo* conservation, the migrations number for chickens is also large enough to reduce the effects of genetic drift.

The paper also needs to expand its description of the populations and how they are managed. A major issue is how populations management is referenced. In the paper these are described as:

- R -ex-situ in-vivo or control around the year 2000. Since this represents 20 years before the experiment I am assuming these samples were cryopreserved.

- O – in situ conservation >20 years, NCF
- A - ex-situ in-vivo >20 years, NCGR

Based upon FAO guidelines the manner in which the populations are referred to is incorrect and therefore leads to confusion in evaluating the results. If samples were collected in ~2000 they would have been cryopreserved or frozen until analyzed, therefore to refer to them as ex-situ in-vivo is incorrect. The only difference between O and A is the physical location, neither (based on Figure 1 map) is where the breeds were developed. Therefore, they both should be designated as ex-situ in-vivo. Making some assumptions about what the authors may have meant the following adjustments are suggested:

- R – Control, cryopreserved/frozen samples,
- O – ex situ in-vivo NCF
- A - ex situ in-vivo NCGR

As such what is being evaluated is how the O-NCF and A-NCGR populations have drifted apart from R.

Response: Special thanks to you for your good suggestions. It is really true as Reviewer suggested that the description of the populations is a bit confusing. For more accurate and easy understanding, we update the nomenclature about different populations as:

- **Con** – cryopreserved samples, as a control
- **In** – *in situ* conservation, NCF
- **Ex** – *ex situ in vivo* conservation, NCGR

The difference between In(O) and Ex(A) is the physical location, while In(O) represents *in situ* conservation, where conserved in NCF (the breeds evolved or are now normally found and bred), and Ex(A) represents *ex situ in vivo* conservation, where conserved in NCGR. As such what is being evaluated is how the In-NCF and Ex-NCGR populations have drifted apart from Con.

The authors should carefully review their statements on LD as they seem to imply that an increase in LD is a positive aspect for conservation rather than a negative.

Response: Thank you very much for your reminding. We are very sorry for our negligence of the statements on LD values. In paper, we observed a higher level of LD in Ex groups of the three breeds compared with In and Con groups. Meanwhile, the homozygosity/heterozygosity (HOM/HET) ratio of different populations exhibited a high degree of consistency with the LD value, with an upward trend in Ex groups and a decrease in In groups. The increase in LD and loss of heterozygosity in Ex groups of the three breeds are likely due to short-term adaptive selection (differences of geographical conditions) or genetic drift (small-population paradigm) during *ex situ in vivo* conservation. Essentially, increase in LD values, loss of heterozygosity and genetic diversity are negative factor for conservation.

We have made correction about the statements on LD in paper.

Reviewer #2 (Remarks to the Author):

This paper reports on a study comparing genomic and methylation changes that occur during conservation of specific indigenous chicken breeds in China. In particular, the authors had a specific interest in the Tibetan chicken, a unique breed native to the Qinghai-Tibet Plateau, which shows distinctive genetic adaptation to high-altitude environments. The aim of this study was to explore this further by comparing the Tibetan chicken to other low-altitude breeds and to identify candidate genomic regions/genes underlying adaptation to high-altitude. This work potentially has significance to conservation programmes and adaptation of specific to environmental changes.

The study raises several interesting observations on genomic variation of genetic and epigenetic characteristics within/between indigenous chicken breeds. Then tries to test for any association (if any) between these measures of genetic and epigenetic variation. Finally, the study explores specific regions/genes for evidence of genetic and/or environmental adaptability to high-altitude. My main concern is that this study is mostly observational and correlates the characteristics of lists of genes highlighted from enrichment analyses from multiple comparisons (between breeds, between conservation programs, etc.), looking for any evidence that supports the role of genes in adaptability towards high-altitude. I would prefer an unbiased, whole genome analysis based on robust statistics. The current study rarely demonstrates this. For example:

- There needs to be a justification of the sampling plan: 3 breeds x 3 conservation regimes x 10 samples each = 90 samples in total – is this sufficient for the comparisons made? (Power Analysis is needed).

Response: Special thanks to you for your good suggestions. For the conservation of animal genetic resources, population size is crucially important. An efficient conservation scheme relies on an effective population size, as well as an effective selection and mating strategy. In China, the government and experts have developed a relatively complete conservation plan (*in situ* conservation and *ex situ in vivo* conservation). Briefly, the conservation goals are that population sizes should be kept constant across generations (30 males and 300 females), and random mating should be enforced within families, with one son kept per sire family and one daughter kept per dam family (Zhang, M. *et al. Genomic diversity dynamics in conserved chicken populations are revealed by genome-wide SNPs. BMC Genomics. 19, 598 (2018).*). Therefore, population size is sufficient and the selection and mating strategy are basically identical whether *in situ* conservation and *ex situ in vivo* conservation. Environmental adaptability is the main factor that determines the difference of chickens under different conservation programmes. We randomly selected 10 individuals from different populations for scientific research. The results were pretty consistent, that the neighbor-joining (NJ) tree featured three groups, with TC, WC, and BC, which distinct accordingly with phenotype and geographic location. Chickens under different conservation programmes basically formed two smaller branches. High consistency across each breed showed a closer genetic affinity within Ex groups and Con groups than that within In groups and Con groups.

Moreover, when the chicken breed was listed in the National Conservation Catalogue and first conserved in the National Chicken Genetics Resources Gene Bank (Jiangsu, NCGR) for *ex situ in vivo* conservation, the migrations number for chickens is also large enough to reduce the effects of genetic drift.

• How robust is the NJ phylogenetic tree? How robust are the branches of the tree (e.g., based on bootstrapping resampling)?

Response: Thank you very much for your reminding. We constructed a neighbor-joining phylogenetic tree based on MEGA X software (v10.1.5) with Kimura 2-parameter model and 1000 bootstrap replicates. The figure of NJ phylogenetic tree with bootstrap values was attached. We also have made correction in Methods.

• What was the statistical measures of significance for the various (LD, Methylation, DMRs, etc.) whole genome analyses?

Response: Thank you very much for your reminding. We have described the statistical measures of significance for the various, such as LD, Methylation, DMRs detection, functional enrichment analysis. See “method” for details.

• The cross-validation analysis of the Structure data was interesting with a minimum error for K= 3, but how significant was this compared to K = 2 or K = 4?

Response: To estimate different ancestral proportions, we used clustering models with Admixture by assuming K ancestral populations. The cross-validation results also demonstrated a clear division between TC, WC and BC at K=3. Then, we calculated the significance of the difference between K= 3 and K = 2/4. The results can be seen in the table below.

groups	p -value
K= 3 vs K = 2	4.552E-16
K= 3 vs K = 4	5.142E-13

• How significant were the overlaps between regions under selective sweeps and DMRs? For example, there are 1000s of DMRs, any overlap could be a sampling artefact (i.e., chance).

Response: To investigate the relationship between genome selection and methylation, we processed the data by overlapping the top F_{ST} region and differentially methylated region, and comparing the region size between *in situ* and *ex situ*. It is unable to perform statistical test on region size data due to the different length of DMRs, so we showed the different proportion of DMRs in F_{ST} and non- F_{ST} region, and we performed biological repeats to avoid sampling artefact.

- How significant were the enrichment analyses for GO and KEGG terms? Were these corrected for multiple testing?

Response: Thank you very much for your reminding. We have described the statistical measures of significance for functional enrichment analysis. The false discovery rate (FDR, q-values) was used to correct the P values with the Benjamini–Hochberg approach. GO terms and pathways with enrichment q-values < 0.05 were considered significantly enriched.

- What is the resolution (and significance) of CGRs/DMRs? How many genes are contained within each region?

Response: DMRs were detected according to the cytosine (C) mapping of uniquely mapped reads into the reference genome. MOABS was used to screen DMRs, and the coverage depth was required to be no less than 10X, with at least three different methylation sites. DNA methylation levels of different groups were compared pairwise through Fisher’s exact test and the p -values were adjusted for multiple comparisons using the Benjamini-Hochberg method. CG-DMR candidate regions were required to have average methylation level differences of > 0.3 between the corresponding populations. Finally, the regions with an FDR less than 0.05 were identified as CGRs/DMRs.

We have added the number of genes within each DMR region in Fig 4a.

Minor but important comments:

- I think English is not the native language of the authors, so I would suggest they seek help with spelling, grammar, and layout of the paper.

Response: Special thanks to you for your good suggestions. We have made completely revised in the paper with spelling, grammar. The layout of the paper has been adjusted at the same time.

- The authors frequently make statements (e.g., methylation is associated with changes in environmental conditions) but do not provide any further explanation or context.

Response: Thank you very much for your reminding. DNA methylation variation levels during *ex situ in vivo* conservation of TC and WC were higher than those during *in situ* conservation. Meanwhile, more DMRs were found during *ex situ in vivo* conservation programme of TC and WC.

As introduced earlier, the environment and climate of the regions of *ex situ in vivo* conservation are remarkably different from those of the origin of indigenous chicken breeds, especially for TC and WC (Fig1a and Supplementary Table 1). Therefore, the differences in climatic and geographic features of the two conservation types can lead to the change in methylation. We have explained it in the revised article.

• Finally, what are “partial candidate genes”?

Response: Thank you very much for your reminding. We have revised it as “candidate genes”.

Reviewer #3 (Remarks to the Author):

This is a well written paper that describes a unique data set which the authors use to assess the impact of ex-situ conservation programs on the genome and epigenome of a model species (chicken). The results are quite interesting and will be relevant for a relatively broad community. I found most of the analyses/results sound (see below for specific queries) and the overall message was quite clear. I have however a few questions for the authors:

Response: Special thanks to you for your good comments.

I suggest re-writing the abstract - it does not really work right now as too many details are given in about the results but there are no hypotheses / questions for example:

Line 10-11: This sentence: " Moreover, a higher proportion of differentially methylated regions in genomic selection regions was found in Tibetan chickens, suggesting a link between the effects of genomic variation and DNA methylation." on its own, makes little sense in the abstract - why specifically Tibetan chickens?

Line 13-14: Without context this sentence " Further analyses revealed that several candidate genes were enriched in the calcium-signaling pathway, which is usually related to high-altitude adaptability " does not make much sense

Response: We have re-written the abstract according to the reviewer’s suggestion. Indeed, we also realized that the abstract was not appropriate, especially in the coherence of the context. We highlighted some commonality results and removed some parts that don't make much sense in the abstract. Details are given in the revised abstract.

line 23: this statement is misleading: this is what the paper cited says: "A molecular clock analysis suggests that domestic chickens diverged from *G. g. spadiceus* ~9500 ± 3300 years ago, though this node does not necessarily correlate with the beginning of domestication process, as chickens are archaeologically visible much later"

Response: Thank you very much for your reminding. We are very sorry for our negligence of the statements on the origin of domestic chicken. Several studies have shown that domestic chickens diverged from the R/JF subspecies *Gallus gallus spadiceus* (Wang et al. 2020 and Peters et al. 2016).

There is also a considerable number of studies given evidences of multispecies origin of domestic chicken (Lawal et al. 2020; Liu et al. 2006; Miao et al. 2013). In order to reduce the misunderstanding to readers, we removed the statement on the origin of domestic chicken, and added some information about indigenous chicken genetic resources by FAO. That would be more in line with the context of this paragraph.

line 45: comparatively to what?

Response: What we meant before is comparatively to other poultry species, such as duck, goose and so on. We both consider that this sentence is not very appropriate. The revised part are as follows:

“Since the genome of RJF has been sequenced and published, the genetic basis of domestication of the chicken has been widely reported in using population genomics.”

Generally please limit the use of acronym to a strict minimum, it is really hard to remember all these acronyms here. I can spot 7 in a single page. Also at least please explain the acronym in the figure legend. Lastly, please use acronym that are more intuitive? Why is A = ex-situ? O = in-situ? why R= control? I am not sure these are going to save that many words? It makes it very difficult to read the figures especially.

Response: Special thanks to you for your good suggestions. For those who first reading this article, it is very difficult as quite a few acronyms. We have rearranged the acronym and deleted considerable words that appear less frequently. Meanwhile, the acronym in all figure legend were explained. For more accurate and easy understanding, we update the nomenclature about different populations as:

- **Con** – cryopreserved samples, as a control
- **In** – *in situ* conservation, NCF
- **Ex** – *ex situ in vivo* conservation, NCGR

I am not sure why the author need to add "in-vivo" to ex-situ?

Response: Add "*in-vivo*" to *ex-situ* is to distinguish from another conservation programme, *ex situ in vitro* conservation.

Conservation activities can be categorized according to whether they involve the maintenance of genetic material *in vivo* or *in vitro*. *In vivo* conservation can, in turn, be classified according to whether it takes place *in situ* or *ex situ*. *In vitro* conservation is conservation under cryogenic conditions including, the cryoconservation of embryos, semen, oocytes, somatic cells or tissues having the potential to reconstitute live animals at a later date.

Thus, conservation actions are commonly grouped into three categories: *in situ* conservation; *ex situ in vivo* conservation; and *ex situ in vitro* conservation.

However, *ex situ in vitro* conservation technology of poultry genetic resource is relatively immature, when compared to another animal genetic resource, such as cattle, sheep, and so on.

Line 101: change "across" to "within" - make it clearer that animals of the same breed still cluster together.

Response: revised as Reviewer suggested.

Line 105-107: I don't think this is necessarily true - could this just be a founder / drift effect? Maybe test this hypothesis using D-statistics? I think this is a very important point here - is the ancestry of these breeds being diluted during conservation? if yes could affect some of the methylation patterns?

Response: Special thanks to you for your good suggestions. We performed D-statistics analysis as follows : P1 : Con-TC and Ex-TC; P2: five branches at the top of In-TC on the phylogenetic tree; P3: bottom five branches of In-TC on the phylogenetic tree; P4: WC. The results showed that Z-score = 11.94 ($P = 1 \times 10^{-16}$). The results further prove that the possibility that other populations could be introduced during *in situ* conservation of TC. The neighbor-joining (NJ) tree and PCA

results both featured three groups, with TC, WC, and BC, which distinct accordingly with phenotype and geographic location (Fig. 1b). Therefore, we think that same breed but other populations was introduced during *in situ* conservation. Actually, this phenomenon happens from time to time during *in situ* conservation which the enterprises are the main body. The variation of methylation patterns is insignificant due to the introduced was the same breed and environment.

Line 117: The result of the increase in LD is very interesting - I would add some more quantitative results in the in text. Also, change decayed to delayed.

Response: Special thanks to you for your good suggestions. Revised as suggested.

Groups	R ² _half	R ² _half_Decay_Value	0.1_Decay_Value
Ex-TC	0.272779132	0.246193513	NA
In-TC	0.244750689	0.215517548	55.68511292
Con-TC	0.260471816	0.210230398	NA
Ex-BC	0.302880734	0.323587138	NA
In-BC	0.297264127	0.455071658	NA
Con-BC	0.28534569	0.276204223	NA
Ex-WC	0.259636559	0.22940031	NA
In-WC	0.244545788	0.220131972	229.9320946
Con-WC	0.247421076	0.196868986	1000.050762

Line 122: how is the hom/het calculated? are the authors considering hom non-ref? or hom-derived allele? or hom sites which are het in one individual? I am also not sure how this can be interpreted - why is this consistent with LD?

Response: Thank you very much for your reminding. If a SNP on homologous chromosomes is the same base, it is called “homozygosity SNP (HOM)”. If a SNP contains a different type of base, it is called “heterozygosity SNP (HET)”. The HOM/HET SNP ratio of different populations exhibited a high degree of consistency with the LD value, with an upward trend in Ex groups and a decrease in In groups when compared with Con groups. Higher ratio of HOM/HET SNP usually indicates lower heterozygosity. Hence, we found an increase in LD values and loss of heterozygosity in Ex groups of the three breeds.

Line 127-128: could this be due to admixture? This would be great to test.

Response: D-statistics analysis above prove that the possibility that other populations of same breed could be introduced during *in situ* conservation. Therefore, the introduced of outgroup results in the increase of genetic diversity during *in situ* conservation of three indigenous chicken breeds.

Line 148-149: Although I agree that some changes might be adaptive, most differences are likely to be the results of a founder / population bottleneck. the authors should generate a few more summary statistics (e.g. Tajima's D?) here which I think could be helpful to see what is going on.

Response: Special thanks to you for your good suggestions. Various indicators of genetic diversity (N_e : effective population size, θ_w : polymorphism and P_N : proportion of polymorphic markers) and Tajima's D during different conservation of three indigenous chicken breeds were assessed. The

average nucleotide diversity (π), effective population size (N_e), polymorphism (θ_w) and proportion of polymorphic markers (P_N) in Ex groups of the three breeds were lower than those in In groups and Con groups (Supplementary Table 4). The values of Tajima's D in all groups were significant deviation from 0, indicating the possibility of natural selection (Supplementary Table 4). High Tajima's D values were found in all three indigenous chicken breeds confirm that there is a bottleneck effect in the evolution of chickens due to the dramatic decrease of population size. Wang et al. also reported that a substantial contraction of the forest area occurred in north equatorial Southeast Asia region during the Last Glacial Period (~125–10 ka), which greatly reduced biodiversity and population sizes of species residing in this area. In our study, Tajima's D values in three indigenous chicken breeds during different conservation programmes did not appear relatively large differences, indicating there was no bottleneck effect happened during the process of conservation. It is really true as Reviewer suggested that founder / drift effect is an important element in conserving *ex-situ in-vivo* populations. However, adaptive changes due to changes in the environment were found, especially in the *ex-situ in-vivo* conservation of TC.

Line 161-164: I think this is a bit strong worded here - it is possible that many (or even all) the differences are due to founder effect rather than adaptation.

Response: Thank you very much for your reminding. It is really true as Reviewer suggested that founder / drift effect is an important element in conserving *ex-situ in-vivo* populations. Currently, *ex situ in vivo* conservation programme of indigenous chicken breeds was inherent to the small-population paradigm, which may increase genetic drift, inbreeding due to low effective population size and demographic isolation. On the other hand, the remarkably change of environment and climate of *ex situ in vivo* conservation can usually lead to adaptive selection, and affect the frequency of alleles. In this paragraph, without considering the influence of other factors, we present some suggestions for conservation programmes according to the physiological characteristics, living environment and risk status of animal resources.

The differentially methylated region analyses are unclear - the authors need to add more information in the method section. What are these differences? Are they between control and ex-situ/in-situ populations? Or are these loci that are differentially methylated within each population? If the latter, are the authors saying that there are more DMRs within ex-situ than in-situ populations? Why would that be? What I think would have been more interesting to look at methylation differences between in-situ and ex-situ to see whether Tibetan population for example shows more differences than other breeds?

Response: Special thanks to you for your good suggestions. We have added the methods (statistical measures of significance, contrasted condition, et. al) of differentially methylated region analyses. In our study, to elucidate the DNA methylation variations occurring during the different conservation programmes of the three indigenous chicken breeds, we identified DMRs between between Con and Ex/In groups (Ex_ and In_). DNA methylation variation levels between Con and Ex of TC and WC were higher than those between Con and In groups. Meanwhile, more DMRs were found in Ex_ group than In_group. As introduced earlier, the environment and climate of the regions of *ex situ in vivo* conservation are remarkably different from those of the origin of indigenous chicken breeds, especially for TC and WC (Fig1a and Supplementary Table 1). Therefore, the differences in climatic and geographic features of the two conservation types

can lead to the change in methylation.

As the reviewer put it, the methylation differences between *ex situ in vivo* (Ex) conservation and *in situ* (In) conservation may have further analytical significance. Therefore, we identified DMRs between Ex and In groups (Ex VS In) of the three indigenous chicken breeds. The results were showed in Fig4b. Compared with In_group (In VS Con), DMRs in Ex VS In group of TC, WC and BC increased by 39.47%, 13.57% and 2.32%, respectively. The results further indicated that the methylation change between *ex situ in vivo* conservation and *in situ* conservation of TC shows more differences than other breeds.

line 249-252 - could you run a test for this maybe?

Response: We obtained candidate divergent regions (CDRs) of the chicken genome between Con and Ex/In groups (Ex_ and In_) of the three chicken breeds by taking a window F_{ST} value of top 5%. DMRs were also identified between Con and Ex/In groups (Ex_ and In_) through the method introduced in the article. Then we took the intersection and complementary set of ranges of two comparison groups through bedtools (2.22) and counted the length of the ranges respectively. The percentage of DMRs in F_{ST} selection regions and non- F_{ST} regions were calculated.

line 259-261: this is weirdly phrased sounds like the authors are saying that DNA methylation creates mutations?

Response: Thank you very much for your reminding. What we actually want to express is that “local genetic variations due to environmental differences during Ex_TC may influence DNA methylation level, and then leads to changes in gene expression”. We have corrected it in the article.

Reviewers' comments:

Reviewer #2 (Remarks to the Author):

I have read the new version of the paper, which has a better layout, reads well and provides more details on the statistical sections I highlighted in my earlier review.

Looks good to go.

Reviewer #3 (Remarks to the Author):

Overall the authors have improved their analyses of differentially methylated regions (DMR) - Figure 4b for example now shows some very interesting patterns.

I am still not convinced by the demographic analyses. For example I am confused by the authors' D-statistics results about the possible introduction of additional individuals into in situ conservation programs. Some of the responses are really unclear (e.g. not sure what "branches at the top" mean) also I cannot see any mention of this analysis in the paper. The authors acknowledge in their response that: "the introduced of outgroup results in the increase of genetic diversity during in situ conservation of three indigenous chicken breeds." Aside from the fact that I do not know what "outgroup" they refer to, I also cannot see any mention of this analysis in the text.

I also cannot see any mention of potential population bottleneck due to founder effect during the establishment of ex-situ population and how this could affect DMRs. It is discussed to some extent in the author's rebuttal but in the text as far as I can see. For example the authors have now computed Tajima's D in all populations, which they say is relatively homogeneous but do not mention the results of this analysis in the text. The discussion of the reason why the authors observe lower genetic diversity in the ex-situ conservation of Tibetan and Wenchang chickens is not convincing - why do the author favour the idea that "other populations of the same breed might be introduced during in situ conservation" over the possibility that there was a founder effect in the ex-situ population?

Dear Reviewer,

Thank you for your comments concerning our manuscript entitled “**Genome and methylation changes in different conservation of indigenous chickens over the last 20 years**”. Those comments are all valuable and very helpful for revising and improving our paper, as well as the important guiding significance to our researches. We have studied comments carefully and have made correction which we hope meet with approval. The main corrections in the paper and the responds to the reviewer’s comments are as following:

Responds to the reviewer’s comments:

Reviewer #3 (Remarks to the Author):

I am still not convinced by the demographic analyses. For example I am confused by the authors’ D-statistics results about the possible introduction of additional individuals into in situ conservation programs. Some of the responses are really unclear (e.g. not sure what “branches at the top” mean) also I cannot see any mention of this analysis in the paper. The authors acknowledge in their response that: “the introduced of outgroup results in the increase of genetic diversity during in situ conservation of three indigenous chicken breeds.” Aside from the fact that I do not know what “outgroup” they refer to, I also cannot see any mention of this analysis in the text.

Response: Thank you very much for your reminding. In the first round of modifications, some points about D-statistics are not clearly responded, and some are not mentioned in the paper. In D-statistics analysis, we used In-WC as as the outgroup and (Con-TC and Ex-TC), (proximal separated clade of In-TC), (distal separated clade of In-TC) as the P1, P2, and P3 groups, respectively. D statistics (Z -score = 11.94) proved that the possibility that other populations could be introduced to In-TC, resulting in a visible genetic differentiation during *in situ* conservation of TC. Actually, this phenomenon that introduced other populations of the same breed to the conservation groups happens from time to time during *in situ* conservation which the enterprises are the main body. The introduced of other populations of the same breed to the conservation groups can lead to the increase of genetic diversity and visible genetic differentiation. We have added D-statistics methods and results, and also relevant discussion in the text.

I also cannot see any mention of potential population bottleneck due to founder effect during the establishment of ex-situ population and how this could affect DMRs. It is discussed to some extent in the author’s rebuttal but in the text as far as I can see. For example the authors have now computed Tajima’s D in all populations, which they say is relatively homogeneous but do not mention the results of this analysis in the text. The discussion of the reason why the authors observe lower genetic diversity in the ex-situ conservation of Tibetan and Wenchang chickens is not convincing - why do the author favour the idea that “other populations of the same breed might be introduced during in situ conservation” over the possibility that there was a founder effect in the ex-situ population?

Response: Special thanks to you for your good comments. In this study, low genetic diversity

occurs during *ex situ in vivo* conservation of three indigenous chicken breeds. We then computed Tajima's D in all populations (Supplementary Table 4). The values of Tajima's D in all groups were significant deviation from 0, indicating the possibility of balancing selection or a founder effect during the process of conservation. Moreover, Tajima's D values of different populations in same breed were relatively homogeneous. These results suggested that environmental adaptive selection may lead to a decline in genetic diversity during *ex situ in vivo* conservation. However, Even so, we cannot rule out the influence of other factors, such as low effective population size and demographic isolation, which is unavoidable limitations during *ex situ in vivo* conservation.

As for the effect of different conservation methods on DMRs, we have identified DMRs between Ex and In groups (Ex VS In) of the three indigenous chicken breeds. The results were showed in Fig4b. Compared with In_group (In VS Con), DMRs in Ex VS In goup of TC, WC and BC increased by 39.47%, 13.57% and 2.32%, respectively. The results further indicated that the methylation change between *ex situ in vivo* conservation and *in situ* conservation of TC shows more differences than other breeds. Then we further analyzed the genes and their possible roles in selective signatures and DMRs, which can provide valuable information about genetic and DNA methylation variations during different conservation programmes.

We have added Tajima's D results and relevant discussion in the text.

And finally, we appreciate you very much for your positive and constructive comments and suggestions on our manuscript. After more than one year of twice revisions, we believe that our work is ready for publication which can take a keen interest for reader about the conservation of genetic diversity for reader.

Thank you and best regards.

Yours sincerely,

Tao Zeng, PhD

E-mail: zengtao4009@126.com

Reviewers' comments:

Reviewer #4 (Remarks to the Author):

Main comments about the revision including D-statistics and response to reviewer #3

The answers to reviewer #3 unfortunately are not written in a clear English. It is difficult for me to understand the meaning of some sentences.

There is also a discrepancy between the informations presented in the response and the text that finally made it into the manuscript. The responses are more detailed and should be reflected better in the manuscript.

l145-160 and methods:Tajima's D: there is no description how the significance was tested.

Furthermore, there is no indication in the main text if Tajima's D was positive or negative. This information needs to be added that the reader can evaluate if balancing selection or a selective sweep might have happened. There is no evidence provided that the deviation from 0 is significant. As a rule of thumb, Tajima's D values can be considered if they are around +2 or -2 (or higher), but the values presented in Suppe. Table 4 are much lower than that.

The authors never mention the possibility of genetic drift as a possible explanation of the observed pattern in their data. In fact, the pattern also could be by chance and not due to environmental adaptive selection. Therefore, the conclusions the authors draw from this analysis about are not justified based on the presented data.

D-statistics:

L100-104 and methods: The sentence is not clear. What is the rationale to use WC as outgroup? It seems that the in situ conserved TC are admixed between WC and BC, so why did the authors use WC as outgroup?

Neither from the methods, nor from the answers to reviewer #3 it is clear what the authors want to test with the D-statistics. As it is written now in lines 100-104, if the aim is to test the ancestral gene flow within the TC (coming from either BC or WC), they need another outgroup - to the best of my knowledge. However, maybe I did not understand correctly the aim of the D3 test. In any case, the authors need to explain it better in the text.

Contrary to what is stated in the abstract, there are no conclusions provided in the manuscript about the practical relevance of the results concerning breed conservation or any recommendations about the "best" conservation strategy.

Figure 1d:

Please re-draw the figure so that the colours for the main groups stay the same throughout the different values of K. E.g., in K=2 the homogenous groups exTC and conTC groups are deep blue, then in K=3 they are red, in K=4 light blue, next red again.

This is much too confusing. Please reformat the figure that those groups, which don't change with different K values keep the same colour. Ideally, the colours can be harmonised with those used in the PCA (1c).

Minor comments:

Check the manuscript for in/ex situ presentation in italics.

l44: Please spell out RJF when it is mentioned for the first time.

l70-73 is redundand with l54-64 and could be removed.

l78-80: The provided percentages roughly sum up to 90%. Where are the remaining 10% SNPs located? How many SNPs were located in exons?

l100: ...showed the possibility that...

l102-104: This sentence does not make sense as it is not written in correct English language. Please rewrite the sentence.

Dear Reviewer,

Thank you for your comments concerning our manuscript entitled “**Genome and methylation changes in different conservation of indigenous chickens over the last 20 years**”. Those comments are all valuable and very helpful for revising and improving our paper, as well as the important guiding significance to our researches. We have studied comments carefully and have made correction which we hope meet with approval. The main corrections in the paper and the responds to the reviewer’s comments are as following:

Responds to the reviewer’s comments:

1145-160 and methods:Tajima's D: there is no description how the significance was tested. Furthermore, there is no indication in the main text if Tajima's D was positive or negative. This information needs to be added that the reader can evaluate if balancing selection or a selective sweep might have happened. There is no evidence provided that the deviation from 0 is significant. As a rule of thumb, Tajima's D values can be considered if they are around +2 or -2 (or higher), but the values presented in Suppe. Table 4 are much lower than that.

Response: In Table 4, several indexes were used to evaluate the genetic diversity during the process of conservation of three indigenous chicken breeds. The values of Tajima's D in nine subgroups were closed to 1(deviation from 0). Just as you said, Tajima's D values can be considered if they are around +2 or -2 (or higher). Based the on the results of Tajima's D, weak balancing selection or a founder effect might have happened during the process of conservation. This is also consistent with the fact that bottleneck effect may happen during the conservation of species in nature. We have added some discussion about Tajima’s D results.

Meanwhile, we did not expanded more about Tajima’s D as our paper was focused on the change of genetic diversity and differentiation during conservation.

The authors never mention the possibility of genetic drift as a possible explanation of the observed pattern in their data. In fact, the pattern also could be by chance and not due to environmental adaptive selection. Therefore, the conclusions the authors draw from this analysis about are not justified based on the presented data.

Response: Special thanks to you for your good suggestions. In our study, we found that low genetic diversity occurs during *ex situ in vivo* conservation. During the process of *ex situ in vivo* conservation, unavoidable limitations such as low effective population size and demographic isolation, which may increase the possibility of genetic drift resulting in a decline in genetic diversity. In addition to the possibility of genetic drift, short-term adaptability to environmental changes may also result in the occurrence of adaptive selection, which can cause a decline in genetic diversity during *ex situ in vivo* conservation.

In contrast to *ex situ in vivo* conservation, chickens that subjected to *in situ* conservation exhibited higher genetic diversity and differentiation, especially in TC and WC. Combined with D-statistics and the actual situation of *in situ* conservation, other populations of the same breed might be introduced during *in situ* conservation, resulting in a increase in genetic diversity.

We have mentioned the possibility of genetic drift as a possible explanation of low genetic diversity during *ex situ in vivo* conservation.

L100-104 and methods: The sentence is not clear. What is the rationale to use WC as outgroup? It seems that the *in situ* conserved TC are admixed between WC and BC, so why did the authors use WC as outgroup?

Neither from the methods, nor from the answers to reviewer #3 it is clear what the authors want to test with the D-statistics. As it is written now in lines 100-104, if the aim is to test the ancestral gene flow within the TC (coming from either BC or WC), they need another outgroup - to the best of my knowledge. However, maybe I did not understand correctly the aim of the D3 test. In any case, the authors need to explain it better in the text.

Response: Thank you very much for your reminding. In our study, two separated clades of In-TC were found in the phylogeny. We then computed D-statistics analysis whether there is gene flow between the two branches of In-TC. Because of visible differentiation were found among these three indigenous chicken breeds (TC\WC\BC). We randomly choose In-WC from WC and BC as the outgroup based on the existing groups to test the gene flow within two separated clades of In-TC. The results (Z-score = 11.94) showed that the possibility that other populations of the same breed could be introduced to In-TC, resulting in a visible genetic differentiation during *in situ* conservation of TC. In fact, during *in situ* conservation, the introduction of other populations of the same breed into the conservation population happens occasionally. Combined with D-statistics and the actual situation of *in situ* conservation, other populations of the same breed might be introduced during *in situ* conservation, resulting in a increase in genetic diversity. Similar results were observed by Wang et al., who showed that TC may have two distinct groups.(Wang, M. S. et al. *Genomic Analyses Reveal Potential Independent Adaptation to High Altitude in Tibetan Chickens. Mol Biol Evol.* 32, 1880-1889 (2015).)

Contrary to what is stated in the abstract, there are no conclusions provided in the manuscript about the practical relevance of the results concerning breed conservation or any recommendations about the "best" conservation strategy.

Response: Thank you very much for your reminding. Our results showed that low levels genetic diversity occurred during *ex situ in vivo* conservation, while higher genetic diversity and differentiation occurred during *in situ* conservation. We should be wary of large degree of genetic differentiation during *in situ* conservation, while the decrease in genetic diversity during *ex situ in vivo* conservation. Therefore, several suggestions may promote better conservation, mainly including 1) In addition to the National Conservation Farm, the Local Genetics Resources Gene Bank (e.g., provincial level, *in situ* conservation) should be established to conserve indigenous chicken resources whose living environment and climate are markedly different from the NCGR, such as Tibetan chicken. 2) Endangered chicken resources (Bian chicken et al.) should be conserved by *ex situ in vivo* conservation (NCGR) to the fullest extent possible because of the limited

population. 3) A combinative conservation programme of *in situ* and *ex situ in vivo*, and regular blood updates, are recommended for normal chicken resources to help maintain high levels of genetic diversity in the long term.

Figure 1d:

Please re-draw the figure so that the colours for the main groups stay the same throughout the different values of K. E.g., in K=2 the homogenous groups exTC and conTC groups are deep blue, then in K=3 they are red, in K=4 light blue, next red again.

This is much too confusing. Please reformat the figure that those groups, which don't change with different K values keep the same colour. Ideally, the colours can be harmonised with those used in the PCA (1c).

Response: Special thanks to you for your good suggestions. We have re-draw this figure.

Minor comments:

Check the manuscript for in/ex situ presentation in italics.

Response: Revised as your suggested.

l44: Please spell out RJF when it is mentioned for the first time.

Response: Thank you very much for your reminding.

l70-73 is redundand with l54-64 and could be removed.

Response: Revised as your suggested.

l78-80: The provided percentages roughly sum up to 90%. Where are the remaining 10% SNPs located? How many SNPs were located in exons?

Response: The remaining 10% SNPs located in CDS, UTR 5', UTR 3', Splice site, Other, et al. Approximately 4% of the SNPs were present in exons.

l100: ...showed the possibility that...

Response: Revised as suggested.

l102-104: This sentence does not make sense as it is not written in correct English language. Please rewrite the sentence.

Response: Thank you very much for your reminding. We have rewrite this sentence.

And finally, we appreciate you very much for your positive and constructive comments and suggestions on our manuscript.

Thank you and best regards.

Yours sincerely,

Reviewers' comments:

Reviewer #4 (Remarks to the Author):

I very much appreciate the efforts of the authors to respond to and implement my comments. E.g. Fig 1d) is much easier to understand now.

I strongly recommend an English editing of the manuscript, as some sentences are not written in correct English grammar.

Unfortunately some issues have not been solved yet:

L70ff: By removing these sentences, the consecutive numbering of the figures is not correct any more.

L78-80: The percentage of functional annotated SNPs is still not clearly described in the text. The authors should present all the percentage numbers (as presented in the rebuttal letter) to sum up to 100%. It is also not clear what is meant with "upstream/downstream gene". Do the authors mean the 5'UTR/3' UTR region of coding region?

Fig 1d) and L93-112

At $K=3$ it becomes clear that the three distinct populations are not TC, BC and WC, but (1) Ex-TC/Con-TC, (2) ExBC/ConBC/InBC and (3) ExWC/ConWC/InWC, while In-TC clearly is an admixed population between EX/Con-TC, BC and WC. Increasing numbers of K confirms that In-TC has shared ancestry with BC and WC.

Therefore it is totally not clear for me, why the authors want to test gene flow within the In-TC "between two branches of In-TC". First, the two branches are not described in the manuscript, and second, there is clear gene flow between In-TC, BC and WC. So it does not make sense to use either BC or WC as outgroup, if the D_3 statistics is used to analyse gene flow within In-TC.

Dear Reviewer,

Thank you for your comments concerning our manuscript entitled “**Genome and methylation changes in different conservation of indigenous chickens over the last 20 years**”. We have studied comments carefully and have made correction which we hope meet with approval. The main corrections in the paper and the responds to the reviewer’s comments are as following:

Responds to the reviewer’s comments:

I very much appreciate the efforts of the authors to respond to and implement my comments. E.g. Fig 1d) is much easier to understand now.

I strongly recommend an English editing of the manuscript, as some sentences are not written in correct English grammar.

Response: Special thanks for your good suggestions. We have carefully corrected any errors in spelling, grammar, and word choice in manuscript by a professional English editing. More details are available in revised manuscript.

L70ff: By removing these sentences, the consecutive numbering of the figures is not correct any more.

Response: Thank you very much for your reminding. Deleted sentences were redundant with the last paragraph of introduction. We have added the numbering of the figures (*Fig. 1a and Supplementary Table 1*) in the introduction.

L78-80: The percentage of functional annotated SNPs is still not clearly described in the text. The authors should present all the percentage numbers (as presented in the rebuttal letter) to sum up to 100%. It is also not clear what is meant with "upstream/downstream gene". Do the authors mean the 5'UTR/3' UTR region of coding region?

Response: Thank you very much for your reminding. We have rearranged the percentage of functional annotated SNPs.

Functional annotation of the SNPs indicated that approximately 61.09% were located in intron, followed by 32.98% in intergenic region, 3.19% in CDS, 2.12% in 3'-UTR, and 0.62% in 5'-UTR.

Fig 1d) and L93-112

At $K=3$ it becomes clear that the three distinct populations are not TC, BC and WC, but (1) Ex-TC/Con-TC, (2) ExBC/ConBC/InBC and (3) ExWC/ConWC/InWC, while In-TC clearly is an admixed population between EX/Con-TC, BC and WC. Increasing numbers of K confirms that In-TC has shared ancestry with BC and WC.

Therefore it is totally not clear for me, why the authors want to test gene flow within the In-TC "between two branches of In-TC". First, the two branches are not described in the manuscript, and second, there is clear gene flow between In-TC, BC and WC. So it does not make sense to use either BC or WC as outgroup, if the D_3 statistics is used to analyse gene flow within In-TC.

Response: Special thanks for your good suggestions. In original version of manuscript, we did not compute D-statistics analysis to test gene flow within the two branches of In-TC. As you have discussed, there is clear gene flow between In-TC and other indigenous chickens (Fig. 1d). Other

populations could be introduced to In-TC, resulting in a visible genetic differentiation during *in situ* conservation of TC. Meanwhile, we did not describe more about the two branches of In-TC as our paper was focused on the overall change of genetic diversity and genetic differentiation during different conservation programmes of three indigenous chicken resources.

In previous rounds of reviews, one of reviewers suggested that this differentiation may be just caused by a founder / drift effect. So, we added D-statistics analysis. In-WC was randomly selected for the outgroup as no better can be choose, to test the gene flow within two separated clades of In-TC. But actually, it is not accurate to use either BC or WC as outgroup. In order to make the method used more reasonable, we all agreed to remove D-statistics analysis in the article.

Finally, we appreciate you very much for your positive and constructive comments and suggestions on our manuscript.

Thank you and best regards.

Yours sincerely.

REVIEWERS' COMMENTS:

Reviewer #4 (Remarks to the Author):

The authors have improved the manuscript again, and I agree with removing the D-statistics. The earlier reviewer's comment about a possible founder effect is very valid. The English language has also been improved. There are several typos/missing spaces left, which the should be corrected during proof reading.

minor suggestions:

l75-77: Functional annotation of the SNPs indicated that approximately 61.09% were located in introns, followed by 32.98% in intergenic regions, 3.19% in CDS, 2.12% in 3'-UTR and 0.62% in 5'-UTR (Supplementary Data 1)

spell out CDS and UTR when used for the first time.

Dear Reviewer,

Thank you for your comments concerning our manuscript entitled “**Analysis of genome and methylation changes in Chinese indigenous chickens over time provides insight into species conservation**”. We have studied comments carefully and have made correction which we hope meet with approval. The main corrections in the paper and the responds to the reviewer’s comments are as following:

Responds to the reviewer’s comments:

Reviewer #4 (Remarks to the Author):

The authors have improved the manuscript again, and I agree with removing the D-statistics. The earlier reviewer's comment about a possible founder effect is very valid.

The English language has also been improved. There are several typos/missing spaces left, which the should be corrected during proof reading.

Response: Thank you very much for your approval. We have made a careful examination of the article about the typos/missing spaces left.

minor suggestions:

175-77: Functional annotation of the SNPs indicated that approximately 61.09% were located in introns, followed by 32.98% in intergenic regions, 3.19% in CDS, 2.12% in 3'-UTR and 0.62% in 5'-UTR (Supplementary Data 2)

spell out CDS and UTR when used for the first time.

Response: Thank you very much for your reminding. We have spell out CDS and UTR when used for the first time.

Finally, we appreciate you very much for your positive and constructive comments and suggestions on our manuscript.

Thank you and best regards.

Yours sincerely.